# Repurposing *α*-Adrenoreceptor Blockers as Promising Anti-Virulence Agents in Gram-Negative Bacteria

**DOI:** 10.3390/antibiotics11020178

**Published:** 2022-01-29

**Authors:** Ahmad J. Almalki, Tarek S. Ibrahim, Sameh S. Elhady, Khaled M. Darwish, Wael A. H. Hegazy

**Affiliations:** 1Department of Pharmaceutical Chemistry, Faculty of Pharmacy, King Abdulaziz University, Jeddah 21589, Saudi Arabia; tmabrahem@kau.edu.sa; 2Department of Natural Products, Faculty of Pharmacy, King Abdulaziz University, Jeddah 21589, Saudi Arabia; ssahmed@kau.edu.sa; 3Department of Medicinal Chemistry, Faculty of Pharmacy, Suez Canal University, Ismailia 41522, Egypt; khaled_darwish@pharm.suez.edu.eg; 4Department of Microbiology and Immunology, Faculty of Pharmacy, Zagazig University, Zagazig 44519, Egypt

**Keywords:** bacterial virulence, quorum sensing, *α*-adrenoreceptor blockers, *Pseudomonas aeruginosa*, terazosin

## Abstract

Antimicrobial resistance is among the world’s most urgent public health problems. Diminishing of the virulence of bacteria is a promising approach to decrease the development of bacterial resistance. Quorum sensing (QS) systems orchestrate the bacterial virulence in inducer–receptors manner. Bacteria can spy on the cells of the host by sensing adrenergic hormones and other neurotransmitters, and in turn, these neurotransmitters can induce bacterial pathogenesis. In this direction, *α*-adrenergic blockers were proposed as an anti-virulence agents through inhibiting the bacterial espionage. The current study aimed to explore the *α*-blockers’ anti-QS activities. Within comprehensive in silico investigation, the binding affinities of seven *α*-adrenoreceptor blockers were evaluated towards structurally different QS receptors. From the best docked *α*-blockers into QS receptors, terazosin was nominated to be subjected for further in vivo and in vitro anti-QS and anti-virulence activities against *Chromobacterium violaceum* and *Pseudomonas aeruginosa*. Terazosin showed a significant ability to diminish the QS-controlled pigment production in *C. violaceum.* Moreover, Terazosin decreased the *P. aeruginosa* biofilm formation and down-regulated its QS-encoding genes. Terazosin protected mice from the *P. aeruginosa* pathogenesis. In conclusion, *α*-adrenergic blockers are proposed as promising anti-virulence agents as they hinder QS receptors and inhibit bacterial espionage.

## 1. Introduction

Drug repurposing is discovering new applications of the well-known drugs which are known for specific clinical use. Re-evaluating the medical use of already-permitted safe drugs has gained increasing interest as a promising strategy that acquires different merits. These advantages include saving time and costs to carry out further pharmaceutical, pharmacological, and toxicological studies for already-approved safe drugs [1,2,3]. There are many examples that prove the efficacy of this strategy and find their way with the new clinical application [1]. Inverse molecular docking is a highly recognized approach that has been widely used to identify multiple putative biological targets, to which small drug-like molecules are able to bind or even weakly bind [4]. The application of this in silico tool has been considered beneficial for facilitating drug design and discovery. A comprehensive inverse docking protocol can be applied for recognizing orphan as well as secondary therapeutic biological targets for drug leads, natural metabolites, and/or any other ligands [5,6]. Additionally, a putative ligand–target inverse docking can predict potential biological targets with possible correlation to the observed drug candidate’s toxicity and side effects [7]. The latter application provides cost-effective and rapid drug testing, particularly within early drug development stages where a clinical candidate could fail out of the clinical trials due to patient intolerance or possessing sever adverse off-target actions [8]. Thus, introducing an inverse docking approach within the drug discovery and development pipeline would speed up the process for rapid market release as well as reduce the cost-of-goods.

Treating antimicrobial-resistant infections or discovering new antimicrobials are among the most valuable aims of drug repurposing which has aroused the curiosity of several research groups [9,10,11,12,13,14,15]. The bacterial capability to develop resistance to all-known antibiotics constitutes a major challenge to the medical teams, patients, and even to the governments [16]. The decreased supply of newly discovered antimicrobials enhanced the need to develop alternative approaches to defeat the bacterial resistance [13,17,18,19]. There are several approaches that have been tested to augment the antimicrobial activity of antibiotics [20,21] or weaken bacterial virulence [14,22,23,24,25]. Bacterial virulence targeting is a promising strategy that confers several advantages; it has no effect on the growth of bacteria, so it does not constitute a stress on bacteria to develop resistance. This strategy does not affect the normal flora as the drugs, natural products, or chemical compounds used at their sub-MIC. Furthermore, the immune system has the chance to complete the eradication of the pathogen [13,17,25,26,27,28]. Markedly, the merits of targeting bacterial virulence are greatly increased when employing safe natural compounds or approved safe drugs [2,12,22,29,30]. In this context, it was suggested that successful quorum-sensing interference (QSI) therapy should not affect bacterial growth or metabolism and is still effective against infectious bacterial strains [31]. Several considerations should be taken in account prior to the use of QSI in clinical applications as an effect on bacterial normal flora [32] and to show if they can be helpful for immunocompromised patients [33].

Bacteria hire a complicated system of inducers and receptors to communicate with each other, named quorum sensing (QS) [34,35]. Professionally, bacterial QS orchestrates bacterial virulence by up-or down regulation of the involved genes during the course of bacterial infection and invasion. Generally, this regulation happens in response to a specific inducer that finds its cognate receptor to form a complex which can bind to the bacterial genome, controlling the expression of specific genes [25,26,34,35]. For instance, Gram-negative bacteria sense inducers are produced by Lux receptors to form a LuxI–R complex which binds with the bacterial DNA at specific sites called lux boxes, which results in a controlled expression of downstream virulence genes [18,34,35]. QS regulates the expression of diverse virulence factors that extend to include bacterial motility, biofilm formation, as well as the production of extracellular enzymes and pigments, as reviewed extensively [25,34,35]. Bearing in mind its crucial role in bacterial virulence, QS is a suitable target to curtail the virulence of bacteria. The present study aimed to repurpose *α*-blockers as anti-QS and anti-virulence agents.

The interkingdom communication between the bacterial and the host cells is important to regulate bacterial infections. In this context, sensors on the membranes of bacteria can spy on the surrounding cells to facilitate their adaptation to the new environment [36]. Interestingly, the autoinducers which are employed in Qs communication systems also cross-talk with the neuroendocrine hormones to activate the same signaling pathway in host cells [37]. This suggests that autoinducers produced by bacteria can enhance the production of adrenaline and noradrenaline, and that bacteria can sense and respond to adrenaline and noradrenaline, enhancing the bacterial virulence, as reviewed [36,38]. Due to the increasing evidence that Gram-negative sense neuroendocrine hormones, it was supposed that blocking the adrenergic receptors, particularly with *α*-blockers, hinders the bacterial receptor-based sensing and mitigates the bacterial virulence [36,37,38,39]. In this direction, there was an aim to evaluate the anti-virulence activities of *α*-blockers, proposing that they may have the ability to block the QS receptors in the same way that they blocked the adrenergic blockers. In this study, an in silico molecular docking study was carried out to investigate the ability of different *α*-blockers to occupy the QS receptors. Then, one of the best docked *α*-blockers into QS receptors terazosin was subjected to further investigation to explore its in vitro and in vivo anti-QS, as well as its anti-virulence activities against *Chromobacterium violaceum* and *Pseudomonas aeruginosa*.

## 2. Results

### 2.1. Double-Staged Multiple Biological Target Docking Investigation

The docking binding affinities of seven FDA-approved *α*-adrenoreceptor blockers against two LuxR-type QSs—*P. aeruginosa*; QscR PDB: 3SZT [40] and *C. violaceum* CviR PDB: 3QP5 [41]—were evaluated. The docking workflow was double-staged. The first stage was a time-saving, rapid preliminary docking protocol for choosing significant leads as compared to the reference co-crystalline ligands of these LuxR-type QS proteins. The next stage is an additional advanced docking approach which aims to more accurately validate the preliminary findings and obtain accurate, valid, and reliable docking poses for the comprehensive evaluation of ligand–protein binding interactions. Validating the second sophisticated docking protocol was proceeded through performing self-docking (redocking) of the co-crystalline ligands by adopting the same docking procedure and, in a fashion, which are comparable to those reported by respected studies [42,43]. Redocked co-crystalline ligands revealed great binding modes of superimposition with significantly low root-mean-square deviation (RMSD) of 1.5834 Å and 1.2416 Å for *C. violaceum* CviR and *P. aeruginosa* QscR pockets, respectively (Appendix A: Appendix A).

Preliminary docking had lower docking energy for QscR co-crystalline control (N-3-oxo-dodecanoyl-_L_-homoserine; O-C12-HSL) than those of CviR’s co-crystalline control (chlorolactone; HLC). The latter co-crystalline inhibitor was assigned with significant docking energies: −7.2051 kcal/mol for *C. violaceum* CviR and −7.6488 kcal/mol at *P. aeruginosa* QscR. These depicted reference binding energies were the cut-offs for identifying promising hits having more negative values. Only three *α*-adrenoreceptor blockers on C. violaceum CviR and two compounds on *P. aeruginosa* QscR showed significantly better docking energies as compared to references (Table 1). Notably, two promising hits—terazosin (Comp. **5**) and prazosin (Comp. **6**)—were satisfactory for the two investigated bacterial LuxR-type QS proteins redeeming further investigation.

Differential docking energy analysis showed that docking energies in all examined ligands were at comparable negative values for both targets. The differential pocket sizes, areas, and volumes across the two target proteins were evaluated via the online Computed Atlas of Surface Topography of proteins server (CASTp; http://sts.bioe.uic.edu/castp; accessed date: 17 September 2021), using a 1.4 Å probe radius under default settings [44]. Richard’s volume solvent-accessible surface area (SASA) were estimated as values of 363.28 Å^3^–516.30 Å^2^ and 331.18 Å^3^–579.64 Å^2^ for the substrate binding sites of *C. violaceum* CviR and *P. aeruginosa* QscR, respectively (Appendix A: Appendix A). The CASTp pocket analysis further demonstrated the differential *C. violaceum* CviR and *P. aeruginosa* QscR pocket topologies. The CviR’s canonical pocket was quite wider, whereas the QscR binding site was more elongated and narrower. It is worth mentioning that the protein–ligand complex interactions were proceeded throughout the second-staged docking protocol.

#### 2.1.1. Analysis of Ligand–QscR *P. aeruginosa* Binding Interactions

Favored ligand–QscR pocket accommodations were prepared for the two preliminary *α*-adrenoreceptor hits due to their relevant contacts with important target’s pocket residues. Common conformational orientation positions were illustrated where the ligand’s aromatic pharmacophoric features were docked at the large-sized hydrophobic sub-pocket, offering minimal steric hinderances (Figure 1). Regarding the ligand’s piperazine-linked *N*-furanoyl heads, significant binding modes of such polar scaffolds were assigned to the QscR’s small sub-pocket. These relevant conformational orientation permitted high-predictable superimposition of the O-C12-HSL, i.e., QscR’s co-crystalline ligand, lactone ring with the *α*-adrenoreceptor, hits polar heads. Notably, extended and elongated conformation was depicted for the docked ligands at the *P. aeruginosa* QscR substrate binding site.

Anchoring the docked ligands at the *P. aeruginosa* QscR pocket highlights the importance of numerous key pocket residues for ligand–protein binding (Table 2). The stability of the docked *α*-adrenoreceptor was mediated via diverse polar amino acids, including Ser38, Tyr58, Trp62, Met127, and Ser129 (Appendix A: Appendix A). Besides the ligand’s hydrophilic binding interactions, these docket hits also illustrated relevant van der Waals non-polar contacts with QscR hydrophobic amino acids, such as Ala41, Tyr52, His53, Tyr58, Trp62, Tyr66, Ile77, Val78, Leu82, Trp90, Phe101, Trp102, Ala105, Ile110, Ile125, Met127, Leu128, and/or Val131. Notably, extended π-associated hydrophobic interactions were also predicted to be relevant for ligand–QscR complex stabilizations. This was most obvious through ligand–Phe54 π–π interaction and ligand–Trp102 CH–π contacts. Both docked hits and reference antagonists depicted additional non-polar van der Waals binding with the side-chain of Arg42 ionizable residue. The above-described residue-wise binding profile was shown comparable to QscR’s potent reference inhibitor. The HLC’s amidic lactone scaffold mediated numerous hydrogen binding with Ser38, Tyr58, Trp62, and Tyr66, while the ligand’s lipophilic aromatic tail showed important hydrophobic links with Phe58 and Trp90 pocket residues. It is worth mentioning that HLC, rather than the *α*-adrenoreceptor hits, managed to depict significant polar interaction with the negatively charged Asp75 sidechain. The latter differential binding modes could be the reason for the particular bulkiness and extended orientation of the docked hits at the *P. aeruginosa* QscR narrow elongated active site.

#### 2.1.2. Analysis of Ligand–CviR *C. violaceum* Binding Interactions

Docking the eight preliminary *α*-adrenoreceptor hits within the *C. violaceum* CviR canonical active site illustrated common conformational orientations. This was best translated into great ligand’s superimposition with comparable pharmacophoric features of co-crystalline ligand, HLC (Figure 2). The ligand’s aromatic rings were oriented towards the large-sized lipophilic sub-pocket at comparable orientation to HLC’s chlorinated phenyl scaffold. On the other hand, the aromatic sulfonamide (Comp. **4**) or saturated/unsaturated furanoyl (Comps. **5**,**6**) heads were directed deep into the CviR’s small sub-pocket, showing great superimposition with HLC’s lactone ring. Unlike the nearly linear conformation of the ligand–QscR binding modes, the docked *α*-adrenoreceptor hits exhibited curved and inverted L-shaped conformation within the *C. violaceum* CviR canonical pocket.

Interestingly, several CviR’s pocket lining residues were accounted for the ligand stability at the designated active site (Table 3). Close-range polar hydrogen bond interactions were mediated via the catalytic negatively charged Asp97 residue only towards the sulfonamide–NH_2_ group of Comp. **4**, where the latter served as a potential hydrogen bond donor. This ligand–Asp97 hydrogen bond pairing was suggested important for anchoring to the ligands at CviR’s small sub-pocket. Further stabilization of investigated hits was associated with extended polar networks towards the wide range of CviR’s pocket residues, such as Tyr80, Trp84, Tyr88, Met89, Asp97, Trp111, and/or Ser115. Only Comp. **6** predicted relevant hydrogen bonding with the NH mainchain of Leu72 via the ligand’s hydroxyl group substituted at the quinazoline ring’s C6 position. In addition to ligand’s polar interactions, almost-conserved van der Waals hydrophobic contacts with Leu57, Ala59, Leu72, Val75, Trp84, Leu85, Tyr88, Met89, Ala94, Pro98, Ile99, Leu100, Trp111, Phe115, Phe126, Ala130, Met135, Ile153, Val250, Met253, and/or Met257 were prophesied for the four investigated hits. Significant π-mediated non-polar interactions were also shown mediated, including π–π interaction with Tyr80, Tyr88, and/or Trp111, as well as close-range CH–π interactions with Leu72, Tyr88, Tyr88, and/or Trp111 sidechains (Appendix A: Appendix A). Only Comps. **4** and **5** illustrated relevant apolar van der Waals binding towards sidechain hydrocarbons (C*β* atoms) of polar residues (Arg42 or Asn92, respectively), which line the CviR’s large hydrophobic sub-pocket. Interestingly, the reference potent inhibitor and CviR co-crystalline ligand, HLC, illustrated comparable residue-wise binding profiles in regard to the investigated *α*-adrenoreceptor hits. The HLC’s amidic lactone drove hydrogen bond pairing with sidechains of Tyr80, Trp84, and Asp97, as well as relevant π-mediated non-polar interactions via its aromatic scaffold towards the Tyr80, Tyr88, and Trp111 sidechains.

### 2.2. Molecular Dynamics Simulation

#### 2.2.1. Analysis of Ligand–QscR *P. aeruginosa* Complex

Monitoring the RMSD deviations of the *P. aeruginosa* QscR proteins in reference to respective alpha carbons (*Cα* RMSD) illustrated typical molecular dynamics (MD) and thermodynamic behaviors (Figure 3A). The protein *Cα* RMSD trajectories elevated over the initial frames following the release of all of the constrains at the start of the runs of MD simulation. Following the initial 20 ns, the three simulated *P. aeruginosa* QscR proteins achieved early convergence and equilibration states for more than half of the MD runs (>140 ns). Limited fluctuations only around 60 ns were represented for the Comp. **5**-bounded *P. aeruginosa* QscR protein, yet these fluctuations never exceeded 0.5 Å above the *Cα* RMSD of the other simulated QscR proteins. The HLC-bounded protein illustrated comparable findings with lower *Cα* RMSD fluctuated tones. Notably, the lowest average QscR protein *Cα* RMSDs were assigned for Comp. **6**-bounded proteins (2.84 ± 0.23 Å), whereas the highest were for HLC and Comp. **5**-bounded ones (3.33 ± 0.34 Å and 3.36 ± 0.33 Å, respectively).

Regarding to the confinement of the simulated ligands at the *P. aeruginosa* QscR substrate binding site, the *Cα* RMSD tones of the sole ligand in relation to the protein’s alpha carbon reference frame were monitored across the 200 ns MD runs. Notably, steady trajectory patterns were illustrated for all simulated ligands (Figure 3B). Ligands managed to attain *Cα* RMSD equilibration plateau with average trajectories ranging from 2.22 ± 0.34 Å to 2.51 ± 0.33 Å, where HLC depicted the highest average value that never exceeded 1.0 Å of any of the simulated ligands. The stability of the ligand’s orientation at the *P. aeruginosa* QscR pocket was further evaluated through the conformational analysis of ligand–protein across the initial and final MD simulation timeframes (0 ns and 200 ns, respectively). Extracted frames of ligand–QscR complexes were minimized to 0.0001 kcal/mol.A^2^ gradient using MOE system preparation software. Limited conformational orientation alterations were shown along these MD simulation timeframes (Figure 3C). The RMSD values of the overlaid frames (aligned RMSD) were 1.980 Å, 1.936 Å, and 2.395 Å for the HLC, Comp. **5**-bound, and Comp. **6**-bound proteins, respectively.

The local residue-wise protein flexibility or fluctuation profiles were investigated to evaluate how this could contribute towards the ligand–QscR protein bindings. The difference root-mean-square fluctuation (ΔRMSF = RMSFapo − holo) was monitored as a function of the *P. aeruginosa* QscR protein residues as a stability validation descriptor. The latter ΔRMSF trajectory analysis was executed along the entire MD simulation timelines since the above-described protein *Cα* RMSDs ensured significant conformational stability among the proteins along the 200 ns for all systems. The ΔRMSF-based mobility cut-off was adopted at 0.03 Å where residues of higher values were considered to have significant immobility and inflexibility. Adopting the latter threshold was reasoned for the ability to identify the immobile residues, while as exclude those that exhibited inherited flexibility, including those with a flexible protein secondary structure (loops) and terminal segments [45,46]. Interestingly, lower immobility and inflexibility patterns were represented for the residue regions vicinal to *N*-terminus rather than the carboxy end, displaying averages of −0.65 ± 0.74 Å and 1.20 ± 0.36 Å, respectively (Figure 4).

Ranges of QscR protein’s core residues had comparable inflexibility and flexibility profiles across the three simulated ligand–QscR *P. aeruginosa* models. Both 74–91 and 111–131 residue ranges depicted high inflexibility profiles with ΔRMSF reaching 1.50 Å. However, the highest stabilized residue region, 180–195, was located vicinal to the carboxy terminus, illustrating the highest immobility profile (ΔRMSF 2.40 Å). On the contrary, the most flexible patterns (ΔRMSF down to highest negative values; −2.50 Å) were assigned to the amino acids along 41–61, 65–75, 135–145, and 165–185 ranges. Trends of more positive ΔRMSFs were assigned for Comp. **5**-bound QscR protein residues as compared to other simulated *α*-adrenoreceptor and reference inhibitors. The latter residue-wise dynamic pattern was most identified for the inflexible regions along 70–90, 105–130, and 180–195 residue ranges within ligand’s binding domain.

Interesting findings were also depicted regarding the specific mobility profiles of the pocket’s key lining residues (Appendix A: Appendix A). Notably, Comp. **5** illustrated the widest range of immobile QscR protein residues, while Comp. **6** shared a comparable pocket residue inflexibility profile as for HLC. The most recognized inflexibility profiles, i.e., ΔRMSF, reaching 1.03 Å, were assigned for several pocket’s residues, such as Ser38, Trp62, Asp75, Ile77, Val78, Leu82, Trp90, Ile125, Met127, and Ser129. Several amino acids (Val78, Trp90, Met127, Ser129, and Ile125) maintained inflexible profile across all simulated ligand–QscR *P. aeruginosa* complexes. Key ligand–pocket binding residues, i.e., Ser38, Trp62, and Trp90, were of relevant inflexibility at the simulated *α*-blocker–QscR protein models.

Exploring the comparative ligand–QscR protein binding affinity, as well as understanding the ligand–protein nature of interaction and individual ligand–residue energy contributions, were pursued through the molecular mechanics and Poisson–Boltzmann surface area (MM/PBSA) free binding energy calculations [47]. This approach is well recognized for having comparable accuracy as the free-energy perturbations approach, however, MM/PBSA offers computational cost effectiveness [48]. Both a single-trajectory approach and an SASA-only apolar solvation model of total free-binding energy calculation (Δ*G*_Total_ = Δ*G*_Molecular Mechanics_ + Δ*G*_Polar_ + Δ*G*_Apolar_) were adopted to correlate high ligand–pocket affinity with depicting more negative binding energy (kJ/mol) values. To our delight, higher Δ*G*_Total_ negative values were allocated for the simulated *α*-adrenoreceptor inhibitors in relation to reference antagonist, HLC (Table 4). Such comparative free energy patterns were in great concordance to the presented docking investigation, showing preferentiality for *α*-adrenoreceptor blockers towards the QscR active site as compared to HLC. Among the investigated ligands, Comp. **5** showed the highest Δ*G*_Total_ with a value of −102.22 ± 6.73 kJ/mol. Comp. **6** showed slightly lower free binding energies (−99.91 ± 10.58 kJ/mol), whereas the HLC was of the lowest ligand–QscR pocket affinity among all simulated ligands.

It is worth mentioning that dominant van der Waal energy contributions (Δ*G*_van der Waal_) were depicted for both the reference and ligand; however, higher values were assigned for Comps. **5** and **6**. Concerning the electrostatic energy contributions (Δ*G*_Electrostatics_), a higher value was seen in the case of Comp. **6**. Notably, both HLC and Comp. **5** showed lower polar solvation energies (Δ*G*_Solvation; Polar_), while almost-comparable Δ*G*_Solvation; Apolar SASA_ was obtained for all ligands. Residues showing favored energy contributions (high negative values) were the active binding site and vicinal residues (Figure 5). Across all simulated ligands, the pocket’s residues, including Phe38, Gly40, Phe54, Try66, Ile77, Val78, and Met127, exhibited the most favored free-binding energy contributions (>−5.00 kJ/mol). On the other hand, moderate energy contributions of ~−3.00 kJ/mol were obtained for Ser38, Ala41, Tyr52, Tyr66, Trp90, Phe101, Ile110, and Ser129. Regarding positive energy contribution, only Lys63 pocket residue for all simulated models inferred repulsion and unfavored role for respective ligand–QscR protein binding. Notably, the catalytic Asp75 showed positive energy contributions with only the simulated *α*-adrenoreceptor blockers yet favored negative energy terms for HLC reference inhibitor.

#### 2.2.2. Ligand–CviR *C. violaceum* Complex Analysis

Typical MD behavior and efficient CviR protein convergence were illustrated through monitoring protein *Cα* RMSDs. Beyond the initial 30 ns, all CviR proteins attained valid convergence and maintained equilibration state till the end of MD runs (Figure 6A). Few initial fluctuations were seen for Comp. **5**, before rapidly attaining its own equilibration. Never exceeding 0.5 Å *Cα* RMSDs above any other simulated models, all simulated CviR proteins managed to converge at near tones (~3.40 Å) at the MD simulation ends. The average protein *Cα* RMSDs were of the lowest tones for Comp. **6**-bound protein (3.05 ± 0.30 Å), while they were the highest for HLC-bound CviR protein (3.70 ± 0.46 Å). Simulated proteins bounded to Comps. **5** and **6** showed the steadiest protein *Cα* RMSDs which follow the initial 30 ns MD run.

Concerning ligand *Cα* RMSDs and ligand–protein pocket confinement, all ligands observed overall steady trajectories (Figure 6B). The average RMSDs were between 2.33 ± 0.22 Å and 2.69 ± 0.39 Å, with the HLC reference ligand depicting the highest *Cα* RMSD tones, yet never exceeding 0.5 Å of any simulated ligands along the whole 200 ns MD simulation timeframes. It should also be considered that both *C. violaceum* CviR and *P. aeruginosa* QscR models depicted comparable ligand *Cα* RMSDs.

Further ligand–CviR active-site orientation stability was illustrated through conformational analysis of ligand–protein models at the initial (0 ns) and final (200 ns) MD timeframes (Figure 6C). Limited conformational orientation alterations were illustrated at these MD simulation runs; however, they were slightly higher instability profiles when compared to QscR findings. The aligned RMSD values of the overlaid frames at 0 ns and 200 ns were 1.949 Å, 2.096 Å, 2.697 Å, and 2.171 Å for the HLC, Comp. **4**-bound, Comp. **5**-bound, and Comp. **6**-bound proteins, respectively.

The ΔRMSF analysis along the 200 ns MD run showed exciting observations. Similar to QscR QS protein, relevant mobility profiles were depicted for the amine end of the proteins as compared to *C*-terminus residues (average −0.44 ± 0.54 Å and 1.15 ± 0.69 Å, respectively) (Figure 7). Nevertheless, the carboxy terminus stability pattern was more profound at *P. aeruginosa* QscR in relation to *C. violaceum* CviR proteins. When obtaining ΔRMSF more than 1.50 Å, high inflexibility patterns were illustrated for 64–81 and 94–116 residue ranges, which were slightly comparable to the core regions of QscR proteins. Additionally, residues along 81–91, 134–141, 151–166, 181–196, and 211–226 ranges were of flexible patterns with ΔRMSF of ~−4.00 Å.

Unlike the above-represented *P. aeruginosa* QscR proteins, an extra residue region (46–56) was depicted with a significant immobile profile (ΔRMSF~1.00 Å) for the simulated *C. violaceum* CviR proteins at a vicinal range towards the *N*-terminus. Comp. **4** depicted the highest negative ΔRMSFs along the 181–196 and 211–226 flexible ranges. Notably, almost all ligand–protein models exhibited comparable ΔRMSF-associated immobility trends across several ligand binding domain regions (Appendix A: Appendix A). Comp. **5** showed the widest residue-wise inflexibility, while the other *α*-adrenoreceptor members depicted higher-pocket residue-associated inflexibility profiles in relation to HLC. Pocket’s amino acids—Leu57, Tyr80, Trp84, Leu85, Met89, Ile99, Leu100, Trp111, Phe115, Ile153, Ser155, and Val250—exhibited the most identified rigid profiles with ΔRMSF reaching nearly 2.17 Å. On the other hand, nine residues (Leu57, Tyr80, Met89, Ile99, Leu100, Trp111, Ile153, Ser155, and Val250) were of consistent inflexibility along the four simulated ligand–CviR *C. violaceum* models. The pivotal pocket–ligand binding amino acids—Leu85, Met89, Trp111, and Phe115—were significantly rigid at proteins in complex with simulated *α*-adrenoreceptor blockers and/or reference inhibitor. Only HLC and Comp. **4** depicted a significant immobility profile for the catalytic Asp79 *C. violaceum* CviR pocket residue.

To our delights, the MM/PBSA free-binding energy calculations of the ligand–CviR *C. violaceum* complexes illustrated higher negative values correlating to more favored binding affinities for simulated *α*-adrenoreceptor blockers as compared to HLC (Table 5). Among the simulated *α*-adrenoreceptor members, Comp. **4** showed the highest comparative total free-binding energy at 137.80 ± 17.97 kJ/mol, while moderate values were obtained for Comps. **5** and **6** (Δ*G*_Total binding_ = −79.49 ± 17.95 and −70.93 ± 29.05 kJ/mol, respectively). These Comps showed better binding affinity towards the *C. violaceum* CviR targets as compared to co-crystalline ligand. Notably, similar ligands (Comp. **5**, Comp. **6**, and HLC) depicted lower Δ*G*_Total binding_ at *C. violaceum* CviR in relation to their respective QscR *P. aeruginosa* complexes.

Dissecting the furnished Δ*G*_Total binding_ of ligand–CviR *C. violaceum* into their contributing energy terms showed dominant Δ*G*_van der Waal_ energy contributions for all simulated ligands, with the highest numbers obtained for Comps. **4** and **5**. However, the electrostatic potential energy contributions were markedly higher for the Comp. **4** protein model reaching up to several folds as compared to those of other simulated ligands. Lastly, lower Δ*G*_Solvation; Polar_ energy terms were assigned for Comp. **6**, as well as the reference control inhibitor. Nonetheless, the Δ*G*_Solvation; Apolar only-SASA_ was almost the same for all ligands. Within a similar fashion to above-described QscR QS complexes, the substrate binding site residues furnished and favored high negative energy binding contributions (Figure 8). With a high binding energy contribution (above −5.00 kJ/mol), the CviR pocket’s lining residues, i.e., Asp97, Ile99, and Leu100, were assumed significant for ligand–CviR model stability. However, Asp97 energy contribution was only significant for Comp. **4** and HLC at values of −6.65 kJ/mol and −3.71 kJ/mol, respectively. The most favored residue-wise energy contribution was obtained for Tyr88 pocket residue (up to −7.31 kJ/mol, 14.25 kJ/mol, −9.11 kJ/mol, and −7.61 kJ/mol for HLC and Comps. **4**, **5**, and **6**, respectively). Moderate energy contributions (around −3.00 kJ/mol) were furnished by Leu72, Val75, Leu85, Met89, Phe126, Ile153, and Ser155 residues. On the contrary, a limited number of polar hydrophobic pocket residues (Arg71, Tyr80, and Thr140) showed unfavored positive contributions to energy for HLC and Comp. **4** (Arg71) along with almost all simulated ligands (Tyr80 and Thr140). These findings infer repulsion effects and an unfavored impact upon the respective stability of ligand–CviR target complexes.

### 2.3. Determination of Minimum Inhibitory Concentration (MIC) of Terazosin against C. violaceum and P. aeruginosa

Based on the in silico docking study of *α*-blockers into the three structurally different QS receptors, terazosin (Comp. **5**) was subject to further investigation to explore its anti-QS and anti-virulence activities. Terazosin inhibited the *C. violaceum* and *P. aeruginosa* growth at 4 and 2 mg/mL, respectively.

To avoid any influence of the terazosin on bacterial growth, it was tested in its sub-MIC (1/4 MIC). Furthermore, the effect of terazosin at sub-MIC on the growth of bacteria was evaluated. Fresh overnight bacterial cultures were inoculated in Luria–Bertani (LB) broth, provided with or without terazosin at sub-MIC. The turbidities of treated or untreated bacterial growth were determined at 600 nm and viably counted. The terazosin at sub-MIC has no significant effect on *C. violaceum or P. aeruginosa* growth (Figure 9). The experiment was performed in triplicate and the results were shown as means ± standard errors. To evaluate the significance, the two-way ANOVA test was used, followed by the Bonferroni post-test. Statistical significance was assumed when *p* values were <0.05.

### 2.4. Terazosin Inhibited the Violacein Production

To preliminary assess the anti-QS activity of terazosin, the biosensor *C. violaceum* is regularly used to evaluate the production of CVi/R QS-controlled violacein pigment [30]. The violacein production was evaluated in the absence or presence of terazosin at sub-MIC. The extracted violacein absorbances were detected at 590 nm. The experiment was conducted in triplicate and the results were expressed as percentage change from untreated control as mean ± standard error. To attest the significance, the Student’s *t*-test was used, and terazosin at sub-MIC significantly reduced the violacein production (*p* < 0.0001) (Figure 10).

### 2.5. Terazosin Downregulated the P. aeruginosa Virulence and QS-Encoding Genes

*P. aeruginosa* is a serious human pathogen; it was chosen to further investigate the anti-QS and anti-virulence activities of terazosin. The inhibitory effect of terazosin at sub-MIC on the expression of genes encoding the three QS systems in *P. aeruginosa* was quantified using RT-PCR (Figure 11). The expressions of QS inducer, receptor-encoding genes (*rhlI/R*, *lasI/R*, and *pqsA/R*), in addition to genes encoding the PmrAB cation-sensing two-component system, were evaluated in *P. aeruginosa* treated with terazosin at sub-MIC, employing the 2^-∆∆Ct^ method. The experiment was reformed in triplicates and the one-way ANOVA test, followed by Dunnett’s multiple comparison test, was employed to attest the significance (where *p* < 0.05 was assumed significant). Terazosin at sub-MIC significantly reduced the expression of tested gens in comparison to the control untreated *P. aeruginosa*.

### 2.6. Terazosin Diminished the Biofilm Formation

To explore the anti-biofilm activity of terazosin, the crystal violet method was used. The absorbance of crystal violet staining the biofilm-forming *P. aeruginosa* in the absence or presence of terazosin at sub-MIC was measured. The experiment was conducted in triplicates and the Student’s *t*-test was employed to demonstrate the significance. The results were shown as percentage change from untreated control as means ± standard errors. Terazosin at sub-MIC significantly diminished the formation of biofilm by *P. aeruginosa* (*p* = 0.0002) (Figure 12).

### 2.7. Terazosin Diminished the P. aeruginosa Motility

Terazosin at sub-MIC significantly diminished the swarming and swimming motilities of *P. aeruginosa* (Figure 13). The experiment was completed in triplicate and the results were shown as means ± standard errors. The Student’s *t*-test was employed to test the significance.

### 2.8. Terazosin Decreased the P. aeruginosa Virulence

*P. aeruginosa* produces various QS-regulated virulence factors to establish its invasion of the host cells [3]. The inhibitory effects of terazosin at sub-MIC on the production of protease, hemolysin, and elastase enzymes, in addition to pyocyanin and oxidative stress resistance, were evaluated (Figure 14). The data were shown as the mean ± standard error of percentage change from the untreated control. The Student’s *t*-test was employed to attest the significance between the treated *P. aeruginosa* and the untreated control.

To evaluate the diminishing activity of terazosin at sub-MIC on the proteolytic activity of *P. aeruginosa*, a casein substrate method was used. Terazosin significantly decreased the protease production (*p* < 0.0001). The hemolysin activity of *P. aeruginosa*, treated or untreated with terazosin at sub-MIC, was evaluated spectrophotometrically. Terazosin significantly diminished the *P. aeruginosa* hemolytic activity (*p* < 0.001). The *P. aeruginosa* elastolytic inhibitory activity of terazosin at sub-MIC was evaluated using the elastin–Congo red method. Terazosin significantly reduced the production of elastase enzyme (*p* < 0.0001). The arsenal of *P. aeruginosa* virulence factors compromises also the QS-controlled cytotoxic pyocyanin pigment which plays an important role in *P. aeruginosa* cytotoxicity [49]. Pyocyanin production was significantly lowered in the presence of terazosin at sub-MIC (*p* < 0.0001). It was documented that QS disruption impairs the oxidative response in *P. aeruginosa* [50]. The ability of terazosin at sub-MIC to reduce the resistance of *P. aeruginosa* to oxidative stress was assessed using the disk assay method. Terazosin significantly lowered the resistance of *P. aeruginosa* to oxidative stress (*p* < 0.0001).

### 2.9. Terazosin Protected Mice against P. aeruginosa

The protective activity of terazosin against *P. aeruginosa* virulence was in vivo estimated. In uninfected or PBS-injected (negative control) groups, all mice survived. In the positive control group where mice were injected with untreated *P. aeruginosa*, only three out of ten mice survived. Meanwhile, terazosin protected seven mice injected with *P. aeruginosa* treated with terazosin at sub-MIC (Figure 15). This finding indicates that the terazosin at sub-MIC significantly diminished the capacity of *P. aeruginosa* to kill mice (*p* = 0.0031) using the log-rank test to assess any trends.

## 3. Discussion

Resistance development to antimicrobials is a global threat which needs overcoming. Diverse approaches have been suggested to conquer the bacterial resistance, including targeting bacterial virulence, which is promising. Targeting bacterial virulence strategy confers multiple benefits, especially if safe drugs are used for this purpose [1,2,11]. Bacterial virulence is mostly orchestrated in response to bacterial community regulations via autoinducer–receptor communication systems called quorum sensing (QS) systems [26,34]. QS systems play a key role in controlling and enhancing bacterial pathogenesis; thus, they are suitable targets to mitigate the bacterial virulence [13,25,35]. In this direction, dozens of studies have been accomplished to screen diverse chemical compounds and drugs prior to being repurposed as anti-virulence agents [1,3,10,11,12,22,23,24,29,30,49]. In the current work, there was an aim to evaluate the anti-QS and anti-virulence activities of *α*-blockers.

An inverse molecular docking study was conducted to evaluate the potential and differential affinities of the seven FDA-approved *α*-adrenoreceptor blockers towards QS protein targets belonging to two high-resistant bacteria species. Crystallographic data regarding the adopted QSs, i.e., *C. violaceum* CviR (~120 kDa) and *P. aeruginosa* QscR (~55 kDa), showed that both protein targets are cyclized homo-dimers resolved at 3.25 Å and 2.55 Å, respectively, together with each respective bound small ligand molecule (Appendix A: Appendix A). Both targets share common architecture comprising the *N*-terminus α/β/α-sandwich domains for ligand binding as well as α/β-loop/α-motifs at the carboxy end which are specifically used for binding to corresponding DNA sites. Binding ligands appear virtually deprived from solvent as being fully embedded at the protein active sites. The latter allows dominant ligand–pocket hydrophobic interactions between the non-polar lining residues at a larger hydrophobic sub-pocket and the ligand’s terminal lipophilic acyl tails and scaffolds. Nevertheless, few hydrogen bond interactions were also depicted for the natural ligand’s lactone heads and amide linkers towards the pocket’s polar residues, permitting selectivity and preferentially deep anchoring at the small-sized inner sub-pocket.

Throughout our presented double-stage docking protocol, both co-crystalline ligands of each QS target (HLC at CviR and O-C12-HSL at QscR) were applied as positive reference controls. Within the current literature, HLC has been widely identified as a potent anti-virulent agent against *C. violaceum*, permitting protection of *Caenorhabditis elegans* against the earlier QS-mediated lethality [51]. Superior antagonism towards CviR *C. violaceum* was also reported by this synthetic small molecule when compared to other investigated inhibitors, including *N*-odecanoyl-_L_-homoserine lactone (C10-HSL) and *N*-octanoyl-_L_-homoserine lactone (C8-HSL), while showing no partial antagonism activity [41]. The profound antagonistic activity of HLC has been identified as efficient against several QS pathway targets as well as across different microbial species, including *Vibrio fischeri* LuxR, *P. aeruginosa* LasR, and *A. tumefaciens* TraR [52]. Here, within the docking protocol, HLC was considered a relevant additional positive control ligand for QscR as an available potent wide-range LuxR-type QS antagonist. In these regards, having an *α*-adrenoreceptor blocker with a higher docking binding energy when compared to both HLC and co-crystalline ligand, while depicting comparable HLC-associated binding mode, would be highly recognized as a potential anti-QS agent possessing potential anti-virulence activity.

Docking protocol validation is important to ensure the credibility of the obtained docking poses and energies, as well as their successful translation into meaningful predicted biological potentialities. Thus, adopting the self-docking (redocking) protocol for co-crystalline ligands was beneficial to validate the trustworthiness of the adopted second stage docking protocol, which was also highlighted in several reported studies [42,43]. As significantly low RMSDs (<2.0 Å) were depicted for the redocked co-crystalline ligands, the adopted parameters of docking and algorithms were confirmed valid and accurate for generating the best docking poses [53]. Importantly, the second-stage docking analysis further validated the preliminary docking results where the investigated hits maintained their higher docking scores in relation to the positive reference ligands. Thus, an investigation of the obtained ligand–QS interactions in correlation with obtained docking scores was of great necessity for providing valuable insights regarding the ligands’ structural aspects which influence the ligand–pocket binding.

The depicted ligand’s conformational orientations at QscR and CviR pockets were highly impacted by the differential size and topology aspects of these active sites. This was highly rationalized since the CASTp pocket analysis illustrated lower accommodation volume and area for the QscR *P. aeruginosa* pocket, while CviR C*. violaceum* depicted a larger pocket size as its ligand was extended towards the solvent side. This was also consistent with current literatures describing the *P. aeruginosa* QscR pocket as having tight packing density based on differential binding modes of variable pheromones [40]. As well as the pocket size and volume differences, the CASTp analysis also revealed topological differences, where the QscR pocket is elongated or narrow rather than wide, as in the case of CviR. This differential pocket topology was further confirmed via the obtained conformational analysis of the docked ligands. Ligands at the QscR pocket adopted almost-linear conformations, while the inverted L-shaped conformations were illustrated for CviR-docked ligands. Despite the differential pocket topology and volume, few *α*-blockers (Comps. **1** and **3**) predicted comparable docking energies for both CviR and QscR targets, whereas the rest of the docked ligands predicted preferential binding energies at either CviR or QscR pockets. Thus, other factors, including ligand topology; the number, type, and nature of ligand substitutions; and hydrophobic–hydrophilic nature of the lining residues of the pockets could have a significant influence on ligand–QS binding interactions as well as pocket accommodations.

Our docking simulation highlighted the significance of polar-driven interactions for anchoring ligands at QscR when compared to CviR targets. This was highly reasoned since hydrophobic interactions were almost conserved for all docked ligands, while more extended hydrophilic networks were depicted with similar ligands binding at the QscR active site. This QscR-specific preferentiality towards polar-directed bindings was also consistent across several reported studies [40,54]. Docking studies further signified the ligand’s preferential binding to particular residues (Ser38), which was also reported as vital for defining the QscR signal specificity through guided preferential binding of native 3-O-HSL over the native unsubstituted ligands [40]. The above findings were reasoned where the investigated high docking scored selective α1-blocker hits (−8.0143 and −8.1023 kcal/mol), showing highly ordered and extended polar contacts with the QscR lining residues. Comp. **5** predicted significant polar interactions, with Ser38, Trp62, Tyr66, Met127, and Ser129 being mediated via its piperazine–quinazoline or furanyl moieties. A comparable pattern of extended polar interactions was shown for Comp. **6**, where Ser38, Tyr58, Trp62, Met127, and Ser129 showed preferential strong hydrogen bond distances and angles. The docking of these ligands was additionally fortified via extended non-polar interactions with aromatic heterocyclic as well as alkyl chain residues. The latter confers an important balance between non-polar and polar interactions for mediating the best docking scores for the ligand–QscR binding. It is also worth mentioning that structural bulkiness and inflexibility could impact ligand–QscR pocket fitting. The possession of terminal-substituted fused heteroaromatic quinazoline ring caused Comps. **5** and **6** to exhibit limited maneuvers to circumvent potential steric hindrances with the bulky residues lining the QscR pocket. Both compounds anchored their saturated or unsaturated furanyl head into the small-sized sub-pocket which can predict the relevant hydrogen bonding with the Trp62 NH ε sidechain. The reasonable flexibility of the central piprazine ring allowed both polar-mediated stabilization of the ligand at the center of QscR pocket, while the substituted quinazoline ring down to the distal hydrophobic pocket was direct. Despite the predicted steric penalties imposed by the quinazoline ring at QscR’s distal pocket, furnishing polar interaction with Met127 mainchain, as well as sidechains of Ser38, Tyr66, and/or Ser129, would compensate the suggested steric penalties, to some extent.

Notably, the above-depicted ligand–QscR binding interactions were found consistent through current literature. Several reports on promising anti-QscR *P. aeruginosa* molecules depicted comparable interaction patterns, as obtained with our docking study. Substituted bis-phenyl hits with central amide linker showed significant *P. aeruginosa* biofilm inhibition activity while predicting polar contacts with Trp62, Tyr66, and/or Asp75, as well as π-driven interactions with Tyr66/Trp90 [55]. Several FDA-approved sulfonamide and carboxamide-based antimicrobial analogues predicted favorable hydrogen bond pairing with Tyr58, Trp62, Asp75, and Ser129, as well as π–π hydrophobic interaction with Tyr66 within computational studies [56]. Finally, both in vitro LasR reporter gene evaluation and in silico investigations revealed potential QscR antagonistic activity of triphenyl-structured compounds, highlighting the ligand–protein binding with Try58, Trp62, Tyr66, Trp63, Asp75, Trp90, Phe101, and/or Ser129 amino acids [57].

Moving towards the docking simulation of *α*-adrenoreceptor blocker at CviR complexes, less steric hindering bindings and more preferable pocket orientations for the bulky ligands, particularly Comp. **4**, were clearly highlighted within our docking study. The latter compound exhibited significantly extended conformation with steric substituted groups at an ortho position to each other on the terminal aromatic rings. This bulky ligand exhibited a higher docking score at CviR in relation to QscR, where it established favored polar interactions with key residues, including catalytic Asp97, Tyr80, and Ser155 at CviR. Despite the bulkiness of the other simulated *α*-blockers (i.e., Comps. **5** and **6**), these ligands managed to achieve higher docking scores as well as free binding energies at the QscR pocket rather than the wider CviR one. This could be reasoned for the rigidity adopted by both ligands, particularly via their fused quinazoline ring, which made them unable to adopt relevant maneuvers to achieve close proximity towards the pocket’s lining residues and subsequent ligand–target binding interactions. The latter was further reasoned where, despite that Comp. **4** exhibiting a comparable extended size, including those of the quinazoline-based drugs, the earlier had flexible ethanolamine aliphatic linker, allowing it to adopt flexible maneuvers for settling its functionalities at close and proper orientations against the pocket’s lining residues. Thus, it was of no surprise that Comp. **4** managed to achieve relevant binding with the QS catalytic Asp75/97 residues through docking and molecular dynamic studies, i.e., the thing that was missed for Comps. **5** and **6** at either CviR or QscR pockets. In these regards, the more preferential Comp. **4** CviR binding was successfully translated into a higher docking score (−8.7628 kcal/mol) and free binding energy (−137.80 ± 17.97 kJ/mol) at CviR as compared to the other investigated drug members.

The credibility of the above residue-wise ligand–CviR binding preferentiality was confirmed where several reported studies showed comparable ligand–residue profiles. Structural-based screening hits of triazole or piperazine-based derivatives presented considerable energy contributions of Met72, Tyr80, Trp84, Leu85, Tyr88, and/or Ser155 for ligand binding [58]. Despite the crucial role of catalytic Asp97 for native ligand binding, this residue showed much lower contributions for ligand stabilization at the *C. violaceum* CviR binding site, which was in great agreement with our in silico findings. Another study showed promising CviR-based anti-virulence biological activity for flavonoid- and chalcone-based hits, predicting favorable polar interactions with Trp84, Asp97, Met135, and Ser155, as well as non-polar contacts to Tyr80, Leu85, Tyr88, Met89, Trp111, Phe115, and Phe126 [59]. Comparable residue-wise bindings were also demonstrated for several synthetic oxazoline/2-imidazoline-based derivatives through molecular docking-coupled dynamics simulations [60]. Similarly, isolates from *Passiflora edulis* showed the relevant accommodation of the CviR binding site which is mediated through balanced hydrophilic–hydrophobic contacts with Ile57, Tyr80, Tyr84, Leu85, Tyr88, Ile99, Trp111, Phe115, Met135, and Ile153 [61].

The validation of our molecular docking findings were further proceeded through 200 ns explicit MD runs which investigated the thermodynamic stability of the top-docked *α*-adrenoreceptor/QS models. Significant global stability and rapid protein dynamic convergence and equilibration were achieved through monitoring *Cα* RMSD trajectories which achieved steady *Cα* RMSD tones for more than 140 ns. Typically, *Cα* RMSD determines the molecular deviation in relation to its designated reference or original structure, providing valuable insights regarding ligand–target stability and MD simulation protocol validity. High-protein RMSDs confer significant conformational changes and instability, and in the case of sole-ligand RMSD, is has been correlated with poor ligand–pocket accommodation [62]. Convergence of the simulated proteins was validated through depicting less than 1.5-fold differences between the ligands and their respective protein *Cα* RMSD tones, following the achievement of thermodynamic equilibration and until the MD runs end. Additionally, the obtained protein RMSDs inferred successful system minimization, thermal equilibration, and relaxation, prior to MD production, which ensured adequacy of 200 ns MD runs, thus needing no further extension. The ligand stability within the pocket was also assured through conformational analysis where limited ligand’s orientation shifts were depicted within any of the simulated QS pockets, yet with preferential conformational orientation stability at QscR.

Moving to another trajectory-based stability indicating parameter, the estimated RMSF analysis highlighted the MD simulation study and adopted protocol validity. Generally, this parameter offers a valuable assessment of the target’s residues dynamic behavior, represented as the fluctuation or flexibility for each protein’s residue in relation to respective reference position across time [63]. Moreover, monitoring the QS residue fluctuations would provide valuable insights regarding the residue-wise contributions at the ligand–target stability. Our depicted immobility preferentiality for QS’s carboxy terminal in relation to *N*-terminus was in good agreement with reported conformational stability analysis within different LuxR-type QS proteins [40,41,64]. Other depicted stability protein’s regions (70–90 and 100–130) has inferred the significant impact of ligand’s binding at these residues ranges. Notably, these residue ranges were reported to possess relatively conserved hydrogen bonding among the constituting residues as well as with various bounded ligands [40,41,59,64,65,66,67]. However, depicting residue flexibility around 130–145 and 170–180 ranges with distances >15 Å far from the QS-active sites conferred the ability of these pockets to accommodate bulkier ligands. Importantly, our RMSF analysis spotlighted several pocket lining residues which were pivotal for ligand–QS binding where these amino acids exhibited high positive ΔRMSF values. The majority of these ligand-conserved rigid residues are hydrophobic, conferring the significant role of the large hydrophobic pocket as well as the ligand’s terminal lipophilic chain for furnishing stabilized ligand–QS models [67]. On the other hand, selected polar pocket residues (serines, catalytic aspartates, and other polar amino acids) showed significant inflexibility, which emphasized the importance of these polar residues to satisfy the ligand’s polar functionality as well as permitting binding selectivity [41].

To our delight, the above-described pocket’s residue-wise inflexibility trends and hydrophobic nature preferentiality were also highlighted at the MM/PBSA-based free-binding energy calculation. The pocket’s lining amino acids as well as vicinal residues of both QS proteins depicted favored energy contributions within the ligand–protein binding energy, implying significant ligand’s confinement at both QS binding sites. Dominance of overall van der Waal energy terms, as well as a depiction of top-energy contribution for hydrophobic residues, illustrated the significant role of hydrophobic interactions for ligand anchoring at both bacterial QS-active sites. This is in a great compliance with the reported data showing the bacterial LuxR-type QSs pockets which are more hydrophobic in nature, deep, and conserved non-polar lining amino acids [40,41,64]. Nevertheless, the presence of polar lining residues would provide reasons to accommodate polar functionalities within the ligands. Thus, the ability of Comp. **4** to depict higher Δ*G*_Electrostatic_ when compared to other *α*-adrenoreceptor blocker agents was reasoned since the earlier compound possesses higher numbers of polar functionalities (hydrogen bond acceptors and donors). The latter polar preferentiality permits Comp. **4** to satisfy the electrostatic potentiality of the polar lining residues at QSs small sub-pocket as well as the terminal part of large-sized hydrophobic subsite. Nonetheless, possessing such polar functionality could be double-bladed as the same ligand depicted high polar solvation energy. It was noting that Δ*G*_Solvation polar_ was lower at CviR as compared to QscR, despite the earlier pocket being more solvent-exposed. The latter was explained due to the tight hydrophobic nature of QscR, allowing the build-up of high-order solvent layers which furnish higher polar solvation energies against ligand–QscR binding. However, the ability of bounded ligands to achieve higher van der Waal potentials at QscR, even higher than at CviR, permitted a compensatory strategy that favored higher ligand binding affinities at QscR in relation to CviR. Based on our in silico study, promising anti-QS *α*-adrenoreceptor blockers would be agents with balanced hydrophobic and polar functionalities at their terminal aromatic scaffold. The latter structural aspect would allow these ligands to furnish favored hydrophobic contacts with QSs large hydrophobic pocket as well as hydrophilic interactions, with polar lining amino acids being vicinal to the small more polar QSs sub-pocket. Such interaction patterns would also reduce any build-up solvation energy penalty which could compromise the process of ligand anchoring.

Based on the presented in silico molecular study, Comp. **4** presented the most superior binding affinity towards CviR, whereas both quinazoline-based ligands showed dual affinity towards both investigated QS targets with higher preferentiality for Comp. **5** over Comp. **6**. Bacterial resistance is considered multi-pathed where the QS-based signaling pathway is considered complex and multi-factorial [68]. Accumulated evidence suggested the preferential usage of multi-target agents for counteracting multi-pathed bacterial resistance with an extra advantage of overall efficacy as a result of their collective synergism on both targets [69]. In these regards, the top dual QS affinity *α*-blocker hit (Comp. **5**; terazosin) was designated to be furtherly investigated for its anti-virulence and anti-QS activities. At the beginning, the MIC of terazosin was determined and the sub-MIC concentration of terazosin, which has no influence on bacterial growth, was used to exclude the effect of terazosin on bacterial growth. A preliminary evaluation for terazosin anti-QS was performed using *C. violaceum*. According to Harrison and Soby, hundreds of studies employed the biosensor *C. violaceum* to evaluate the Gram-negative QS systems because of its ability to produce the violacein pigment in response to acyl homoserine lactones under the control of CViI/R QS system [70]. Therefore, we quantified the ability of *C. violaceum* to produce violacein in the absence or presence of terazosin at sub-MIC. In agreement with the docking results which explored the terazosin ability to occupy *C. violaceum* CViR, terazosin significantly diminished the production of the QS-controlled pigment violacein. For further examination of the anti-virulence and anti-QS activities of terazosin, it was subjected to further investigations on a more clinically important Gram-negative bacterial model for *P. aeruginosa*.

*P. aeruginosa* is a known abundant human pathogen among the ESKAPE serious pathogens list (*Enterococcus faecium*, *Staphylococcus aureus*, *Klebsiella pneumoniae*, *Acinetobacter baumannii*, *P. aeruginosa*, and *Enterobacter* spp.) [49]. Practically, *P. aeruginosa* can infect all the body tissues, causing diverse acute and chronic infections, as previously reviewed [71,72]. *P. aeruginosa* possesses diverse bundles of virulence factors expanding from biofilm formation; extracellular enzymes production as protease, elastase, and hemolysins; and production of cytotoxic pyocyanin pigment [13,49,72]. *P. aeruginosa* utilizes three QS systems to control its virulence. Two LUX homolog QS systems, namely LasI/R and RhlI/R, sense the C12-homoserine lactone and butanoyl homoserine lactone, respectively [3,13,25]. In addition to the non-Lux QS system, PQS is expressed by PqsA, B, C, D, and H under regulation of PqsR [3,73]. An additional orphan homolog for LuxR “QscR” that binds to the LasI autoinducers [35,40]. The molecular docking study revealed the high ability of terazosin to bind and compete on QscR. Moreover, terazosin at sub-MIC downregulated the expression of the three *P. aeruginosa* QS-encoding genes (lasI/R, rhlI/R, and pqsA/R). In great agreement with the in silico and genotypic findings, terazosin at sub-MIC showed a significant ability to diminish all the tested QS-controlled phenotypic virulence factors. Terazosin at sub-MIC markedly reduced the biofilm formation; motility; and production of cytotoxic pyocyanin pigment and virulence enzymes hemolysins, protease, and elastase.

There is growing evidence of the interkingdom speech, in which bacteria can spy on the host cell to establish its accommodation. Simply, bacteria, particularly Gram-negative bacteria, utilize membranal sensor kinases to sense the adrenergic hormones noradrenaline and adrenaline that results in enhancing the bacterial virulence as reviewed [38,74]. The most studied sensor kinases QseC and E in *E. coli* and *Salmonella* spp. can sense adrenaline and noradrenaline to increase the bacterial resistance [38,75,76]. Surprisingly, *α*-blockers showed a considerable ability to hinder these sensor kinases and diminish the bacterial virulence [38,74,75,77]. These observations establish an additional proposed mechanism of the anti-virulence activity of *α*-blockers, blocking bacterial QS systems and reducing the espionage of bacteria on the host cells.

The membrane-embedded QseC sensor kinase forms a two-component regulatory circuit with QseB in different Gram-negative bacteria that senses adrenergic hormones. It was shown that PmrA and PmrB in *P. aeruginosa* are homologs to QseC and QseB, respectively [78]. It has been posited that *α*-blockers could also hinder PmrAB regulatory system as they block QseC/B [75,77,78]. Moreover, as a consequence, we hypothesized that blocking or downregulating PmrAB in *P. aeruginosa* could diminish its virulence. The *P. aeruginosa* PmrAB is a two-component regulatory system that controls resistance to cationic antimicrobial peptides (CAPs) and aminoarabinose modification of lipopolysaccharides [79,80]. CAPs are short peptides and constitute a major component of innate immunity that encounter the invading pathogens at the surface of epithelial cells [81,82]. The antimicrobial activity of CAPs is attributed to detergent-like nature which enables CAPs to bind to the lipopolysaccharide (LPS) on the surface of Gram-negative bacterial cells that ends by disruption of cell membrane and cell death [17,80]. Importantly, antimicrobial peptides enhance the tolerance to oxidative stress [83]; these findings are in great compliances with our results. In our study, the *α*-blocker terazosin downregulated the expression of PmrAB-encoding genes and significantly decreased the *P. aeruginosa* resistance to oxidative stress. Moreover, CAPs are associated with the *P. aeruginosa* resistance to antibiotics, as the invading *P. aeruginosa* may encounter exogenous CAPs, such as polymyxins. The resistance to polymyxins is associated with the ability of *P. aeruginosa* to produce proteases that can degrade CAPs [84,85]. As a result, the terazosin downregulation to PmrAB system that controls the resistance to CAPs could increase the *P. aeruginosa* susceptibility to antibiotics and decrease its resistance to oxidative stress in immune cells.

A conclusive experiment reflects the anti-virulence activity of the *α*-adrenergic blocker terazosin; it significantly protected mice from the *P. aeruginosa* pathogenesis. Our findings clearly showed the anti-virulence activity of terazosin as an example of *α*-adrenergic blocker. This significant terazosin anti-virulence activity could be attributed to its ability to block QS receptors and membranal adrenergic receptors, in addition to its ability to diminish the bacterial resistance to killing inside immune cells. In this work, we explored the possibility for repurposing of *α*-adrenergic blockers as efficient anti-virulence agents that can be used solely or as adjuvants beside traditional antibiotics. The clinical use of such compounds as *α*-adrenergic blockers needs further pharmacological and toxicological studies, besides an evaluation of the efficiency of different pharmaceutical formulations. This study is a preliminary investigation that opens the way for further detailed pharmacological and pharmaceutical studies to repurpose *α*-adrenergic blockers and similar chemical moieties.

## 4. Materials and Methods

### 4.1. Structural Preparation of QS Targets and Building Ligands for In Silico Investigation

Using MOE-2019 software (CCG^®^; Montreal; Canada), the two QS targets (*P. aeruginosa* QscR PDB; 3SZT) and *C. violaceum* CviR PDB; 3QP5) were prepared through three-dimensional protonation, atom types and connectivity auto-correction, and partial charge assignment at physiological neutral pH 7.4 [86,87]. The investigated *α*-adrenoreceptor inhibitors, as well as control antagonists, were built via MOE module using respective SMILES line annotations. Ligands were then energy-minimized across a gradient-conjugated method (2000 steps till reaching 1 × 10^−3^ kcal/mol/Å^2^ RMS gradient convergence under MMFF94s forcefield and partial charges) [88,89,90].

### 4.2. Double-Staged Multiple Biological Target Docking Investigation

The MOE Alpha Site Geometrical Finder was used to define the potential QS binding site. The canonical binding site of each target was selected from a list of putative active sites obtained from the MOE Alpha Site Finder module (Appendix A: Appendix A). Each target’s canonical binding site was selected with guidance of reported studies [91,92]. Potential binding site were identified based on the MOE site scoring function, indicating the propensity for ligand binding score depending on the composition of the contact residues within the target pocket [93]. Site refinement was then proceeded through matching the MOE-obtained site with that obtained from the CASTp online server, the inclusion of the crystalline ligand, as well as crucial residues reported within the current literature [40,41]. Defined by the geometrical descriptor *alpha spheres*, the pocket sizes of the selected canonical binding site for CviR and QscR QS were 93 and 83 *alpha spheres*, respectively [94]. Lining residues comprising the QS-identified canonical pockets are illustrated within the Appendix A (Appendix A).

Molecular docking was carried out with preliminary fast rigid screening docking protocol for identifying potentials hits showing higher negative docking energies (kcal/mol), as compared to the control antagonist(s). Within the latter stage, residues of the designated QS proteins were maintained inflexible while the ligand’s conformation were generated via ligand placement technique and bond rotation at the defined binding site, guided by a triangular matcher approach [95]. Obtained poses were ranked by London/dG scoring. Following the preliminary stage, a second (more sophisticated) docking process was proceeded for refining the first identified hits to increase the validity and prediction accuracy of the adopted molecular docking protocol. Accounting for QS residue flexibility, the adopted induced-fit docking protocol retained 10 poses following initial triangular matcher and London/dG scoring for subsequent energy minimization and refinement. Sidechains of the QS proteins were set to be tethered within the force field configuration options. Obtained poses were then rescored via GBVI_WSA/dG forcefield by relying on explicit solvation electrostatic, currently loaded charge, exposure-weighted surface area, and protein–ligand van der Waals and Coulombic electrostatics [96,97]. The selection of best ligand–protein was a combined approach of high docking energy, low RMSDs at a 2.0 Å threshold in relation to the co-crystalline ligand, as well as obtaining crucial interactions with literature on pocket residues which are crucial for ligand–QS binding.

PyMol2.0.6 Graphical Software (Schrödinger^®^, New York, NY, USA) was used for visual inspection and ligand–protein interaction analysis [98]. Hydrogen bond cut-off values (Acceptor…H-Donor) were set at a 20° bond angle and a 3.3 Å distance, correlating to optimal hydrogen bond strength [45]. Both the MOE–ligand interactions wizard and Pymol bond distance–angle measurement wizards were used for assessing the potential hydrophobic interactions, keeping a 5.0 Å cut-off distance measured from the residue’s α-carbon atoms to the nearest interacting ligand’s atom.

### 4.3. Molecular Dynamics Simulations

Bacterial QS proteins in complex with their respective promising *α*-adrenoreceptor hits as well as HLC were used as starting coordinate models for a 200 ns explicit molecular dynamics study. The free-license GROMACS_2019 software was used under a CHARMM-General forcefield program and CHARMM36m forcefield for simulating the ligands and proteins, respectively, within TIP3P cubic box at implemented 10 Å marginal distance and periodic boundary conditions [99,100]. Standard ionization states of QS protein residues were set at pH 7.4, and the entire system was neutralized via chloride and potassium ions [101].

Each ligand–QS system was minimized via a steepest descent algorithm for 0.005 ns and then equilibrated for 0.1 ns under NVT-ensemble (303.15 K; Berendsen temperature approach), followed by another 0.1 ns equilibration under an NPT ensemble (303.15 K, 1 atm pressure; Parrinello–Rahman barostat approach). Throughout the minimization and equilibration stages, the 1000 kJ/mol.nm^2^ force constant was applied to preserve original folding of the proteins as well as restrain all heavy atoms. Finally, the restrains were removed at a production stage where systems run for 200 ns under NPT ensemble were proceeded. A particle-mesh Ewald algorithm was applied to compute electrostatic interactions of long ranges [102]. Modelling all covalent bond lengths was achieved via a linear-constraint LINCS technique at a 2 fs integration time-step [103]. The Verlet cut-off scheme was used for truncating van der Waals and Coulomb’s non-bonded interactions at 10 Å [104].

Trajectory analysis was performed using GROMACS built-in scripts for RMSD and RMSF, while the latter was best represented in difference expression (ΔRMSF) between the each holo/liganded QS protein and its apo/unliganded state (i.e., RMSF_apo–holo_). The Apo QS proteins were prepared, minimized/equilibrated, and finally simulated under similar conductions as that of the holo QS proteins. The GROMACS “*g_mmpbsa*” module was implemented to calculate the free binding energy of investigated hits/reference towards QS protein via the MM/PBSA calculation [48]. This free-energy calculation was applied on representative frames from the entire simulation runs (200 ns). Representing the ligand–protein conformational analysis at designated timeframes was achieved using the PyMol2.0.6 software.

### 4.4. Bacterial Strains Chemicals and Microbiological Media

*P. aeruginosa* PAO1 (ATCC BAA-47-B1) and *C. violaceum* CV026 (ATCC 31532) were employed in this study. Terazosin *α*-blockers (CAS numbers: 63074-08-8) was purchased from Sigma-Aldrich (St. Louis, MO, USA). Luria–Bertani (LB) broth and agar, tryptone soya broth (TSB), and Mueller–Hinton (MH) broth and agar were obtained from Oxoid (Hampshire, UK). The chemicals were of pharmaceutical grade.

### 4.5. Determination of MIC of Terazosin and Its Effect at Sub-MIC on the Growth of Bacteria

An agar dilution method was used to determine the MIC of terazosin, according to the protocol of Clinical and Laboratory Standards Institute (CLSI, 2015) [2,3]. To avoid any influence of terazosin on the growth of *C. violaceum* or *P. aeruginosa*, the bacterial growth in the presence of terazosin at sub-MIC (1/4 MIC) was evaluated [12,30]. Briefly, fresh overnight bacterial cultures were cultured in LB broth with or without terazosin at sub-MIC at 37 °C overnight. The bacterial cultures’ turbidities were detected at 600 nm. For more confirmation, viable counts (CFU/mL) were performed for bacterial cultures.

### 4.6. Evaluation of Violacein Production in C. violaceum

The terazosin ability to diminish the QS-controlled violacein production by *C. violaceum* was evaluated as previously described [2,30]. Briefly, aliquots of LB broth with the N-hexanoyl homoserine lactone autoinducer in the absence or presence of terazosin at-sub-MIC were mixed with equal volumes of *C. violaceum* suspensions (O.D_600_ 1) in microtiter plates and incubated at room temperature overnight. Then, plates are left to dry and the violacein pigment was extracted with dimethylsulfoxide (DMSO) and the absorbances were detected at 590 nm. The results were calculated as a percentage change from the untreated negative control cultures.

### 4.7. Quantitative RT-PCR of P. aeruginosa Virulence Involved and QS-Encoding Genes

A quantitative real-time PCR was performed as described previously [11,105]. The RNA of *P. aeruginosa*, treated or untreated with terazosin at sub-MIC, were extracted using the provided protocol by the Purification Kit Gene JET RNA (Thermo Fisher Scientific, Waltham, MA, USA). The expression of *P. aeruginosa* QS-encoding genes *pqsA*, *pqsR*, *rhlI*, *rhlR*, *lasI*, and *lasR*, as well as PmrAB system-encoding genes *pmrA* and *pmrB*, were quantified in the presence and absence of terazosin at sub-MIC by qRT-PCR. The relative levels of genes’ expressions were normalized to the *ropD* expression level as a housekeeping gene. The used primers were listed in (Appendix A: Appendix A). The extracted RNA was used to synthesis cDNA using a high-capacity cDNA Reverse Transcriptase Kit (Applied Biosystem, Carlsbad, CA, USA). The Syber Green I PCR Master Kit (Fermentas, Waltham, MA, USA) was used to amplify the cDNA in the StepOne instrument (Applied Biosystem, Carlsbad, CA, USA). The relative expression of tested genes was detected by the comparative threshold cycle (∆∆Ct) method [9,10].

### 4.8. Evaluation of P. aeruginosa Biofilm Formation

The crystal violet method was used to assay the effect of terazosin at-sub-MIC on the biofilm production in *P. aeruginosa* as previously described [24,106]. *P. aeruginosa* (1 × 10^6^ CFU/mL) 100 µL LB broth aliquots in the absence or presence of terazosin at-sub-MIC were transferred to microtiter plates and incubated at 37 °C for 24 h. Then, planktonic cells were washed out, and adhered cells were fixed with 99% methanol for 20 min and stained with 1% crystal violet for 25 min. Excess dye was washed out and plates were left to dry. Crystal violet was extracted with 33% glacial acetic acid and the absorbances were detected at 590 nm. The biofilm formation was considered as a percentage change from untreated cultures. To visualize the biofilms’ formation, the biofilms were left to be formed on cove slips in the absence or presence of terazosin at sub-MIC.

### 4.9. Evaluation of P. aeruginosa Motilities

The *P. aeruginosa* swimming or swarming was evaluated in the absence or presence of terazosin at sub-MIC as described earlier [3,30]. Briefly, standard 5 µL *P. aeruginosa* inoculums (1 × 10^6^ CFU/mL) were centrally inoculated in 0.3% or 0.5% LB agar plates provided with or without terazosin at sub-MIC for swimming or swarming motility assay, respectively. After overnight incubation, the zones of motility were measured in mm.

### 4.10. Evaluation of P. aeruginosa Hemolysins

The anti-hemolytic activity of terazosin at sub-MIC was evaluated as previously described [12,23]. Briefly, the supernatants were collected from *P. aeruginosa* cultures treated or not with terazosin at sub-MIC. Equal volumes of supernatants were mixed with fresh 2% erythrocytes suspensions, incubated at 37 °C for 2 h, and centrifuged. At the same conditions, negative control of un-hemolyzed erythrocytes and positive control were prepared by adding 0.1% sodium dodecyl sulfate (SDS) to an erythrocyte suspension. The absorbances were detected at 540 nm, and the hemolysis in the presence or absence of terazosin was compared to positive and negative controls. The results of treated *P. aeruginosa* were evaluated as a percentage change from the hemolysis of untreated control cultures.

### 4.11. Evaluation of P. aeruginosa Protease

The anti-proteolytic activity terazosin at sub-MIC against *P. aeruginosa* was evaluated using the skim milk agar method [23,24]. Briefly, the supernatants were collected from *P. aeruginosa* cultures treated or not with terazosin at sub-MIC. Equal volumes of supernatants collected from treated or untreated *P. aeruginosa* were moved to the wells in 5% skim milk agar plates and incubated at 37 °C overnight. The diameters of clear zones due to protease were measured in mm. The results obtained by treated *P. aeruginosa* were evaluated as a percentage change from those obtained by untreated bacterial controls.

### 4.12. Evaluation of P. aeruginosa Elastase

The anti-elastolytic activity of terazosin at sub-MIC against *P. aeruginosa* was assessed using the Congo red method [3,30]. The reagent elastin Congo red (ECR) was prepared with 10 mg of ECR in 500 µL buffer (0.1 mol/L Tris pH 7.2 and 10 mol/L CaCl2). The supernatants were collected from *P. aeruginosa* cultures treated or not with terazosin at sub-MIC. Supernatants were mixed with the ECR reagent, incubated 37 °C for 6 h, and then centrifuged to remove insoluble reagent. The absorbances were measured at 495 nm. The elastolytic activity of terazosin-treated cultures was evaluated as percentage change from untreated bacterial culture.

### 4.13. Evaluation of P. aeruginosa Pigment Pyocyanin

The inhibitory effect of terazosin at sub-MIC on formation of the *P. aeruginosa* pyocyanin pigment was assessed as described earlier [30,49]. Briefly, standard *P. aeruginosa* overnight cultures were inoculated in 1 mL of LB broth with or without terazosin at sub-MIC, incubated at 37 °C for 48 h, and centrifuged. The absorbances of the produced pyocyanin were measured at 691 nm. The terazosin inhibitory effect on the production of pyocyanin was calculated as a percentage change from untreated bacterial culture.

### 4.14. Evaluation of P. aeruginosa Resistance to Oxidative Stress

The disk assay method was employed to evaluate the effect of terazosin at sub-MIC on the *P. aeruginosa* tolerance to oxidative stress [3]. Briefly, standard *P. aeruginosa* cultures were spread uniformly on the surface of LB agar plates, provided with or without terazosin at sub-MIC. Then, 10 μL of hydrogen peroxide (1.5%) was added on the sterile paper disks placed on the prepared LB agar plates. After overnight culture at 37 °C, the inhibition zones of *P. aeruginosa*, treated or untreated with terazosin, were measured in mm.

### 4.15. In Vivo Protection Assay

To assess the in vivo anti-virulence activities of terazosin, the mice survival model was used to evaluate the terazosin’s in vivo protective activity against *P. aeruginosa*, as previously described [23,49]. Briefly, fresh overnight *P. aeruginosa* cultures in LB broth, with or not terazosin at sub-MIC, were adjusted to approximately 1 × 10^6^ CFU/mL) in phosphate-buffered saline (PBS). Three-week-old female *Mus musculus* mice were distributed in 4 groups (*n* = 10). The test group was intraperitoneally (ip) injected with 100 μL of terazosin-treated *P. aeruginosa* in sterile PBS. Two negative control groups were uninfected or were injected with sterile PBS. The positive control group was injected with untreated *P. aeruginosa*. The survival of mice was observed over 5 successive days and plotted using Kaplan–Meier method.

## 5. Conclusions

The development of bacterial resistance is an endless battle that mandates a dynamic development of new approaches and strategies. Repurposing safe and approved drugs to curtail bacterial virulence is one of the promising approaches and acquires multiple advantages. In this study, we aimed to repurpose *α*-adrenergic blockers as anti-virulence agents. Based on the outcome of the in silico study, terazosin as a top dual QS affinity *α*-adrenoreceptor hit was identified, which is promising for further testing its anti-virulence activities. Terazosin significantly mitigated the QS-controlled virulence factors in *C. violaceum* and *P. aeruginosa*. Terazosin downregulated QS-encoding genes and protected mice against *P. aeruginosa*. This study shows the terazosin anti-virulence activity is owed to its anti-QS effects besides its blockade to bacterial espionage on the host cells and decreasing bacterial resistance to killing inside immune cells. In this work, the anti-virulence activities of *α*-adrenergic blockers were preliminary explored, thus keeping the door open for further detailed investigation prior to their clinical application.

## Figures and Tables

**Figure 1 antibiotics-11-00178-f001:**
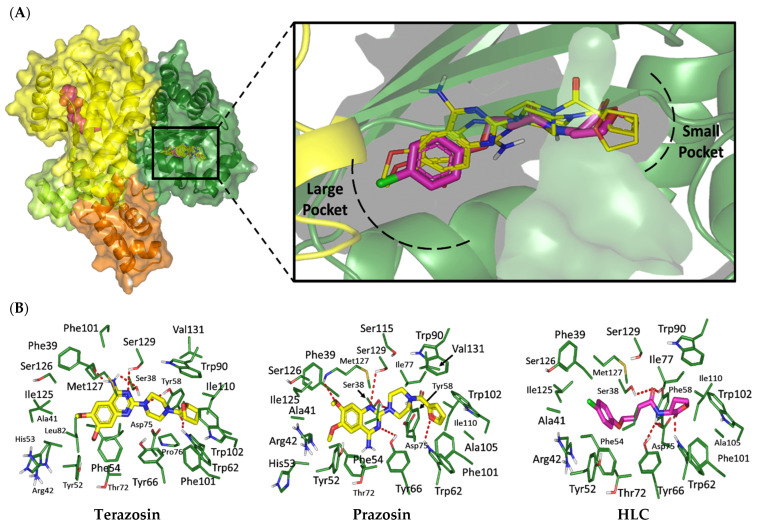
Molecular binding interaction of the ligand–protein complexes. (**A**) Cartoon and surface representation of *P. aeruginosa* QscR (PDB; 3SZT), showing an overlay of investigated *α*-adrenoreceptor hits (yellow lines) over reported reference inhibitor, HLC (magenta sticks), at the QS protein’s substrate binding site of protomer-B (green). The binding site involves the small (more polar) and larger (more hydrophobic) sub-pockets. At protomer-A (yellow), the O-C12-HSL co-crystalline ligand is represented as magenta spheres. (**B**) Predicted docking poses of the examined ligands (sticks). Residues within 5 Å radius of the in complex ligands were only displayed, colored based on their respective Qs subsite location, and sequentially labeled with numbers. For clarity, non-polar hydrogens are removed, while hydrogen bonding was represented as red dashed lines.

**Figure 2 antibiotics-11-00178-f002:**
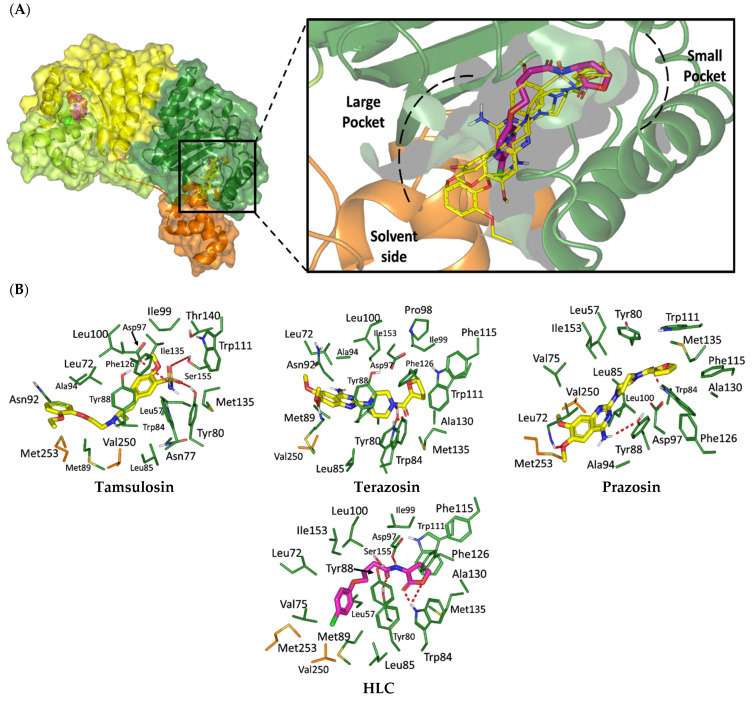
Molecular binding interaction of the ligand–protein complexes. (**A**) Cartoon and surface representation of *C. violaceum* CviR (PDB; 3QP5), showing an overlay of investigated *α*-adrenoreceptor hits (yellow lines) over reported reference inhibitor, HLC (magenta sticks), at the QS protein’s substrate binding site of protomer B (green). The binding site involves the small (more polar) and larger (more hydrophobic) sub-pockets. At protomer A (yellow), the co-crystalline ligand, HLC, is represented as magenta spheres. (**B**) Predicted docking poses of the investigated ligands (sticks). Residues within 5Å radius of the in complex ligands were only displayed, colored based on their respective Qs subsite location, and labeled with sequence number. For clarity, non-polar hydrogens were removed, while hydrogen bonding was depicted as red dashed lines.

**Figure 3 antibiotics-11-00178-f003:**
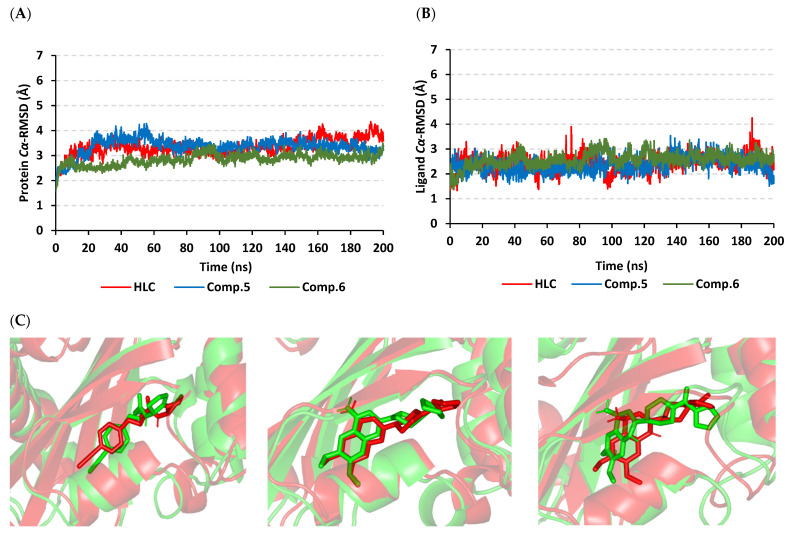
Analysis of ligand–QscR *P. aeruginosa* model stability across the 200 ns explicit MD simulations. Alpha carbon RMSD (Å) trajectories of (**A**) *P. aeruginosa* QscR proteins; (**B**) *α*-adrenoreceptor and control inhibitors, against MD simulation time (ns). (**C**) Overlaid frames of the ligand–QscR *P. aeruginosa* protein complexes at initial (0 ns) and end (200 ns) of MD runs. The left, middle, and right panels are for HLC, Comp. **5**, and Comp. **6** protein complexes, respectively. Both the ligands (sticks) and *P. aeruginosa* QscR proteins (cartoon) are illustrated in green or red, corresponding to the initial and end extracted frames, respectively.

**Figure 4 antibiotics-11-00178-f004:**
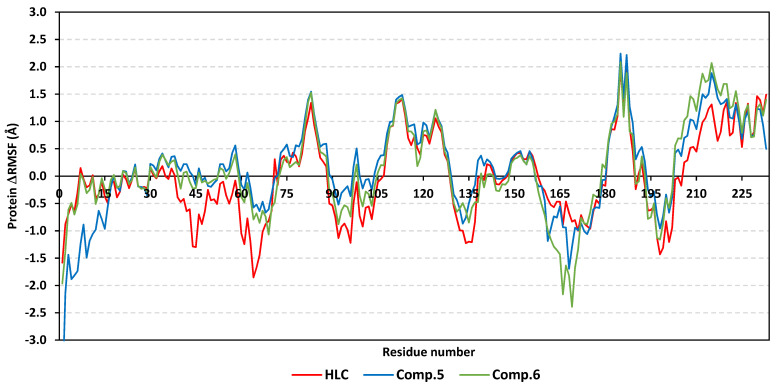
Residue–wise global stability analysis of *P. aeruginosa* QscR proteins in relation to ΔRMSF trajectories across the entire 200 ns MD simulation timeframes. The estimated ΔRMSFs are illustrated as a function of the *P. aeruginosa* QscR protein residue numbers, where the latter are calculated assuming independent MD simulation of *P. aeruginosa* QscR apo/non-liganded states against the complexed holo proteins in bound to *α*-adrenoreceptor or reference inhibitors.

**Figure 5 antibiotics-11-00178-f005:**
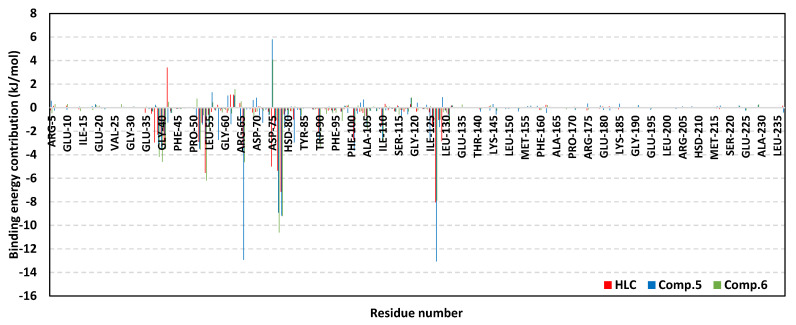
Residue–wise binding energy contributions within the total free–binding energy calculation of ligand–QscR *P. aeruginosa* complexes. Binding energy contributions are illustrated as a function of QscR *P. aeruginosa* protein residue numbers.

**Figure 6 antibiotics-11-00178-f006:**
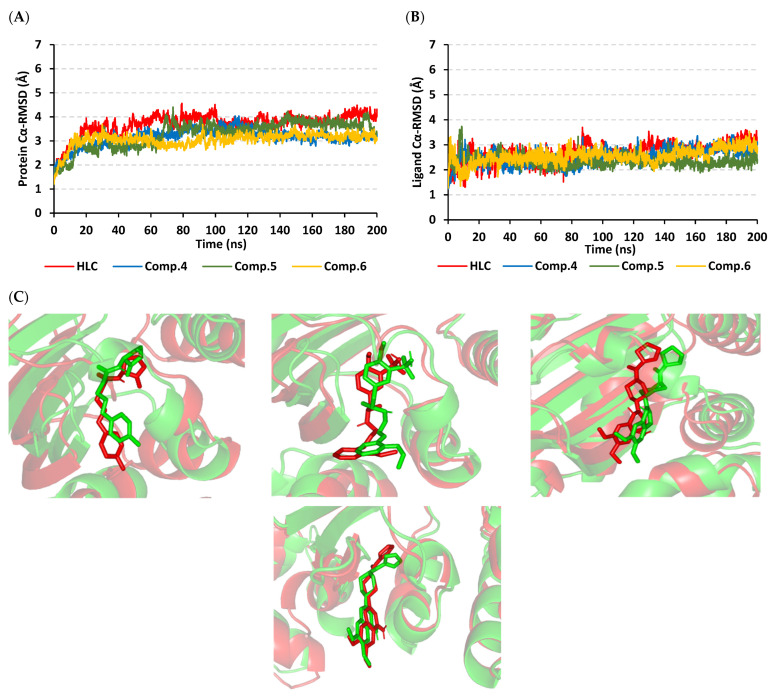
Analysis of ligand–CviR *C. violaceum* model stability across the 200 ns explicit MD simulations. Alpha carbon RMSD (Å) trajectories of (**A**) CviR *C. violaceum* proteins; (**B**) *α*-adrenoreceptor and control inhibitors, against MD simulation time (ns). (**C**) Overlaid frames of the ligand–CviR *C. violaceum* protein complexes at the start (0 ns) and end (200 ns) of MD runs. The left, middle, and right panels are for HLC, Comp. **4**, Comp. **5**, and Comp. **6** protein complexes, respectively. Both the ligands (sticks) and *C. violaceum* CviR proteins (cartoon) are illustrated in green or red, corresponding to the start and end of the extracted frames, respectively.

**Figure 7 antibiotics-11-00178-f007:**
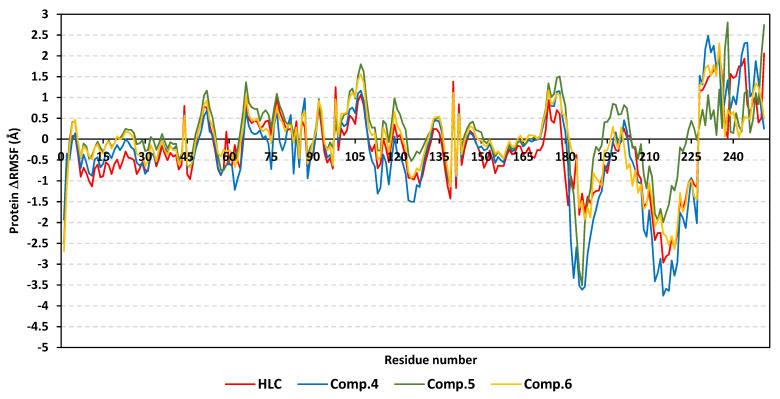
Residue–wise global stability analysis of *C. violaceum* CviR proteins in relation to ΔRMSF trajectories across the entire 200 ns MD simulation timeframes. The estimated ΔRMSFs are illustrated as a function of the *C. violaceum* CviR protein residue numbers, where the latter are calculated considering independent MD simulation of *C. violaceum* CviR apo/non-liganded states against the complexed/holo proteins in bound to *α*-adrenoreceptor or reference inhibitors.

**Figure 8 antibiotics-11-00178-f008:**
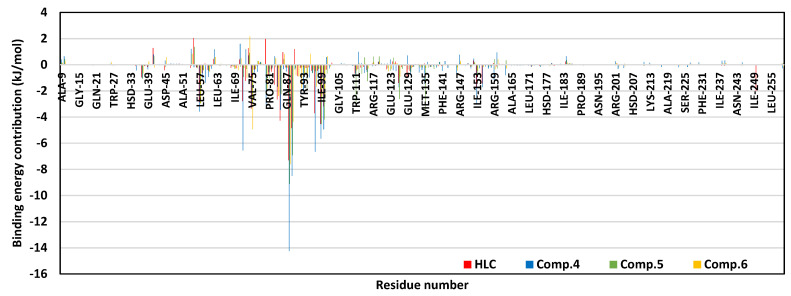
Residue–wise binding energy contributions within the total free–binding energy calculation of ligand–CviR *C. violaceum* complexes. Binding energy contributions are illustrated as a function of CviR *C. violaceum* protein residue numbers.

**Figure 9 antibiotics-11-00178-f009:**
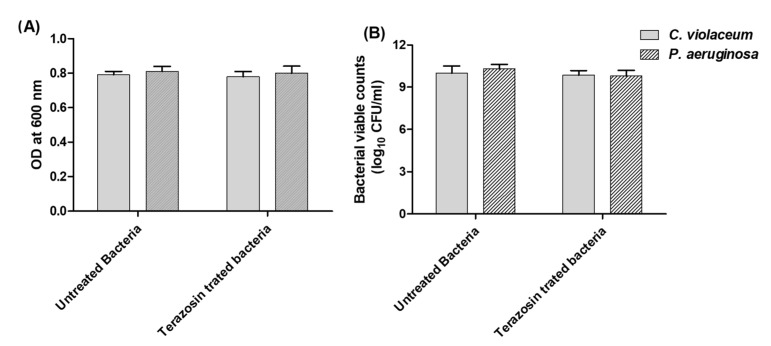
Effect of terazosin on *C. violaceum* or *P. aeruginosa* growth. (**A**) The bacterial growth turbidities in the presence or absence of 1/4 MIC of terazosin were measured at OD 600 nm. (**B**) Viable count of terazosin-treated and control untreated bacterial cultures after overnight incubation. There was no significant effect of terazosin at sub-MIC on *C. violaceum* or *P. aeruginosa* growth.

**Figure 10 antibiotics-11-00178-f010:**
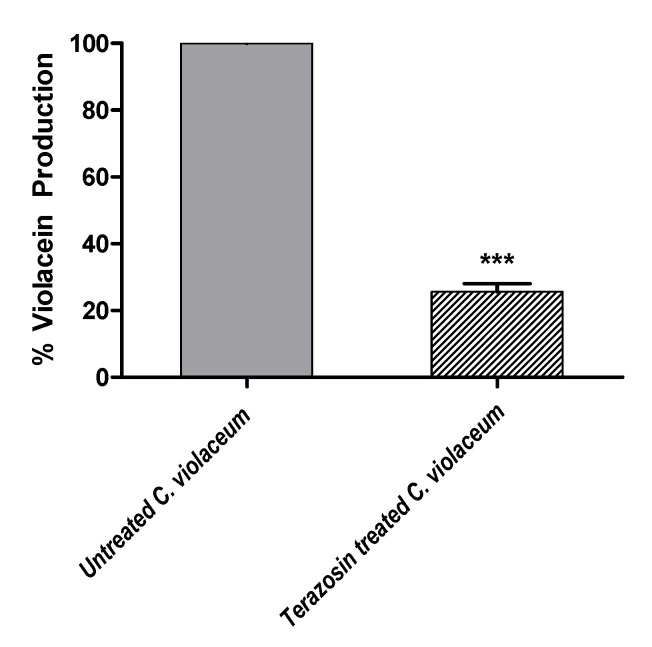
Terazosin at sub-MIC effect on the production of QS-controlled violacein pigment. *C. violaceum* CV026 was allowed to grow in the absence or presence of terazosin at sub-MIC. The produced violacein was extracted by DMSO and the absorbances were evaluated at 590 nm. Terazosin significantly diminished the production of violacein (***: *p* ≤ 0.001).

**Figure 11 antibiotics-11-00178-f011:**
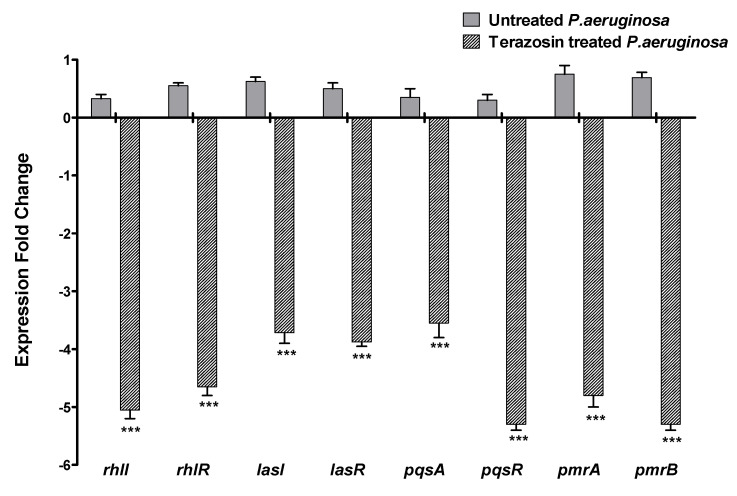
Terazosin decreased the expression of virulence and QS genes of *P. aeruginosa*. RNA of *P. aeruginosa*, treated or not with terazosin at sub–MIC, was isolated, and the expression of each gene was normalized to the housekeeping gene *rplU* gene. The test was performed in triplicate and the results were expressed as mean ± standard error. The one-way ANOVA test, followed by Dunnett’s multiple comparison test, was used for statistical analysis. Terazosin significantly downregulated the expressions of all tested genes, ***: *p* ≤ 0.001.

**Figure 12 antibiotics-11-00178-f012:**
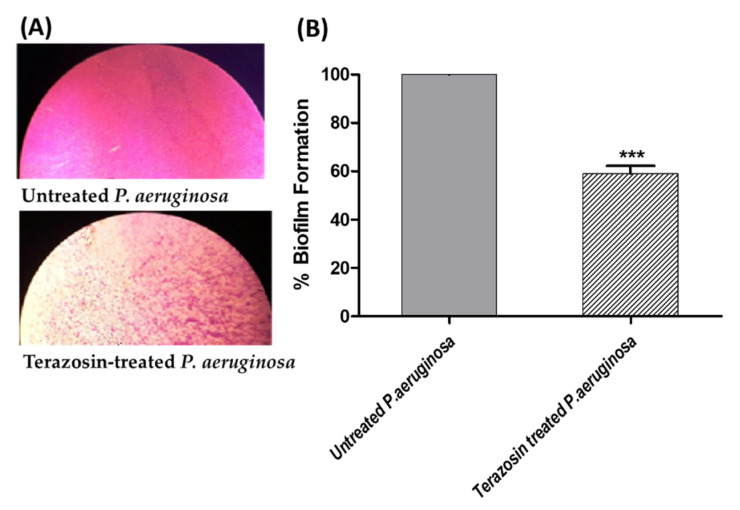
Terazosin diminished the formation of biofilm in *P. aeruginosa*. A crystal violet method was used to stain the biofilm-forming cells in the absence or presence of terazosin at sub-MIC. (**A**) Light microscopic images showed a few dispersed adhered cells when treated with terazosin at sub-MIC. (**B**) The absorbance of the crystal violet staining the biofilm-forming cells in the absence or presence of terazosin at sub-MIC was measured at 590 nm. The results are presented as the percentage change from the untreated *P. aeruginosa* control. The Student’s *t*-test was employed to test the significance; terazosin significantly reduced biofilm formation, ***: *p* ≤ 0.001.

**Figure 13 antibiotics-11-00178-f013:**
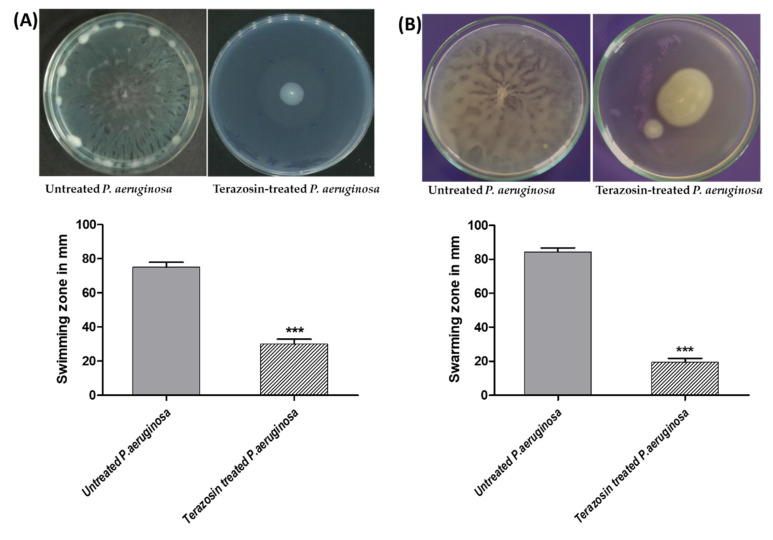
Terazosin diminished the (**A**) swarming and (**B**) swimming of *P. aeruginosa* motility. The swarming or swimming motility zones were measured in the absence or presence of terazosin at sub-MIC. The test was repeated in triplicate. The Student’s *t*-test was used to test the significance. Terazosin significantly diminished the *P. aeruginosa* motility, ***: *p* ≤ 0.001.

**Figure 14 antibiotics-11-00178-f014:**
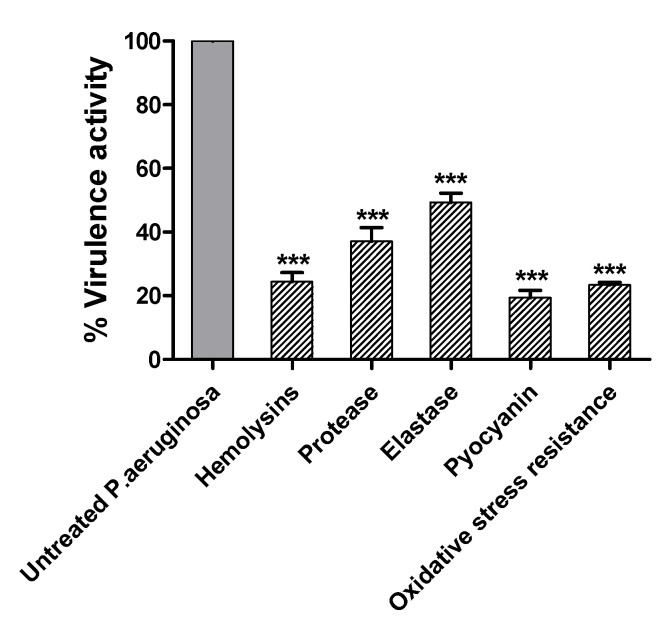
Terazosin decreased the *P. aeruginosa* virulence. Terazosin at sub-MIC significantly reduced the production of hemolysins, protease, and elastase enzymes, as well as pyocyanin pigment, and the resistance to oxidative stress. The results were presented as the percentage change from the untreated *P. aeruginosa* control. The experiments were completed in triplicates and the Student’s *t*-test was employed to attest the significance between treated *P. aeruginosa* and untreated control, ***: *p* ≤ 0.001.

**Figure 15 antibiotics-11-00178-f015:**
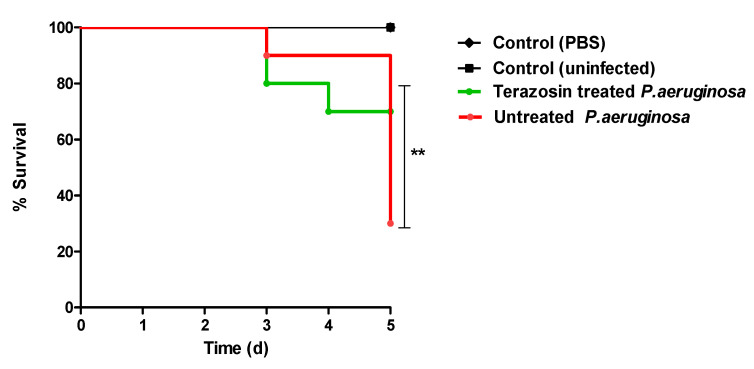
In vivo protection from *P. aeruginosa*. Four groups composed of ten healthy female mice were recruited. In the negative control groups, mice were intraperitoneally injected with PBS or left un-injected. In the positive control group, mice were injected with untreated *P. aeruginosa.* The last group was injected with *P. aeruginosa* treated with terazosin at sub-MIC. The mice survival in different groups was observed over 5 days and the deaths were plotted using the Kaplan–Meier method. The significance (*p* < 0.05) was attested using the log-rank test. In the negative control group, all mice survived. However, terazosin at sub-MIC protected seven mice from death, in comparison to three mice which survived in the positive control group. Terazosin showed significant reduction in the *P. aeruginosa* capacity to kill mice (the log-rank test was used to assess the trends: *p* = 0.0031). **: *p* ≤ 0.01.

**Table 1 antibiotics-11-00178-t001:** Docking binding energy for the FDA-approved *α*-adrenorecptor inhibitors and co-crystallized reference ligands towards two bacterial LuxR-type QS (CviR *C. violaceum* PDB: 3QP5/QscR *P. aeruginosa* PDB: 3SZT) across the preliminary docking investigation.

Compound	2D Structure	Nomenclature	Docking Binding Energy (Kcal/mol) ^a^	
			3SZT	3QP5
**1**	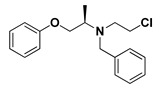	Phenoxybenzamine	−6.8144	−6.4353
**2**	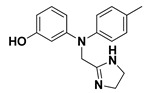	Phentolamine	−4.7680	−5.5767
**3**	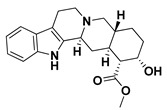	Yohimbine	−4.2864	−4.0237
**4**	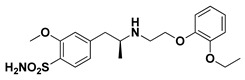	Tamsulosin	−6.9369	**−7.8553**
**5**	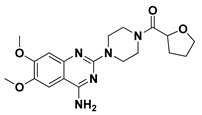	Terazosin	**−7.4416**	**−7.5163**
**6**	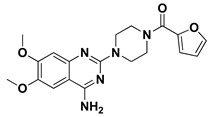	Prazosin	**−7.5679**	**−7.2600**
**7**	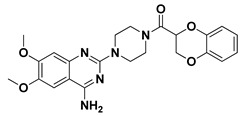	Doxazosin	−6.6578	−7.0163
**3SZT Reference**	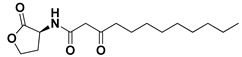	**O-C12-HSL**	−7.5547	-
**3QP5 Reference**	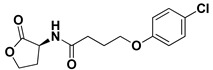	**HLC**	−7.6488	−7.2051

^a^ MOE-developed docking binding energies obtained from triangular matcher and London_dG scoring for the investigated *α*-adrenoreceptor blockers in relation to reference ligand. Higher negative values are in bold and highlighted.

**Table 2 antibiotics-11-00178-t002:** Descriptors of the ligand–QS protein docked complex at *P. aeruginosa* QscR (PDB; 3SZT) binding site throughout the secondary flexible-based docking protocol.

Compound	Docking Binding Energy (kcal/mol) ^a^		H-Bond (Polar) Interactions	Hydrophobic Contacts	π-Interactions	van der Waal with C-Side Chain
	Preliminary (Rigid)	Secondary (Induced-Fit)				
**Terazosin**	−7.4416	−8.0143	Ser38, Tyr58, Trp62, Met127, Ser129	Ala41, Tyr52, His53, Tyr58, Trp62, Tyr66, Ile77, Val78, Trp90, Phe101, Trp102, Ile110, Ile125, Met127, Val131	Phe54 (π–H) Trp102 (π–H)	Arg42 (C*β*)
**Prazosin**	−7.5679	−8.1023	Ser38, Tyr58, Trp62, Met127, Ser129	Ala41, Tyr52, His53, Tyr58, Trp62, Tyr66, Ile77, Val78, Leu82, Trp90, Phe101, Trp102, Ala105, Ile110, Ile125, Met127, Leu128, Val131	Phe54 (π–π) Trp102 (π–H)	Arg42 (C*β*)
**HLC**	−7.6488	−7.9912	Ser38, Tyr58, Trp62, Tyr66, Asp75	Phe39, Ala41, Tyr52, Tyr58, Trp62, Ile77, Val78, Phe101, Trp102, Ala105, Ile110, Ile125, Met127	Phe54 (π–π) Trp90 (π–H)	Arg42 (C*β*)

^a^ MOE-developed docking binding energies based on triangular matcher and London_dG first scoring, followed by refinement and second scoring via GBVI/WSA_dG forcefield rescoring function.

**Table 3 antibiotics-11-00178-t003:** Descriptors of the ligand–QS protein docked complex at the *C. violaceum* CviR (PDB; 3QP5) binding site throughout the secondary flexible-based docking protocol.

Compound	Docking Binding Energy (kcal/mol) ^a^		H-Bond (Polar) Interactions	Hydrophobic Contacts	Π Interactions	van der Waal with C-Side Chain
	Preliminary (Rigid)	Secondary (Induced-Fit)				
**Tamsulosin**	−7.8553	−8.7628	Tyr80, Asp97, Ser115	Leu57, Leu72, Val75, Trp84, Leu85, Met89, Ile99, Leu100, Trp111, Phe126, Met135, Ile153, Val250, Met253, Met257	Tyr80 (π–π) Tyr88 (π–H)	Asn92 (C*β*)
**Terazosin**	−7.5163	−8.3934	Tyr80, Met89, Trp84, Trp111	Leu57, Ala59, Leu72, Val75, Trp84, Leu85, Tyr88, Met89, Ala94, Pro98, Ile99, Leu100, Phe115, Phe126, Ala130, Met135, Ile153, Val250, Met253	Tyr80 (π–H) Trp111 (π–H)	Arg42 (C*β*)
**Prazosin**	−7.2600	−8.0092	Leu72, Trp84, Tyr88	Leu57, Leu72, Val75, Trp84, Leu85, Ala94, Ile99, Leu100, Phe115, Phe126, Ala130, Met135, Ile153, Val250, Met253	Leu72 (π–H) Tyr80 (π–H) Tyr88 (π–π) Trp111 (π–π)	-
**HLC**	−7.2051	−8.08374	Tyr80, Trp84 *, Asp97	Leu57, Leu72, Val75, Trp84, Leu85, Met89, Ala94, Ile99, Leu100, Phe115, Phe126, Ala130, Met135, Ile153, Val250, Met253	Tyr80 (π–H) Tyr88 (π–π) Trp111 (π–H)	-

^a^ MOE-developed docking binding energies based on triangular matcher and London_dG first scoring, followed by refinement and second scoring via GBVI/WSA_dG forcefield rescoring function; * signifies multiple polar interactions for the designated QS residue towards the ligands.

**Table 4 antibiotics-11-00178-t004:** Total free-binding energies (Δ*G*_Total binding_) and contributing energy terms (Δ*G*_X_) for the *α*-adrenoreceptor inhibitor hits as well as reference antagonist towards the *P. aeruginosa* QscR substrate binding site.

Energy (kJ/mol ± SD)	Ligand–Protein Complexes		
	HLC	Comp. 5	Comp. 6
**Δ*G*_van der Waal_**	−122.79 ± 14.13	−241.58 ± 15.41	−239.11 ± 2.66
**Δ*G*_Electrostatics_**	−46.75 ± 2.55	−56.77 ± 2.86	−72.40 ± 2.83
**Δ*G*_Solvation; Polar_**	120.42 ± 1.28	217.32 ± 4.88	233.61 ± 17.03
**Δ*G*_Solvation;_** ** _Apolar; Only-SASA_ **	−18.75 ± 0.04	−21.19 ± 0.95	−22.01 ± 0.97
**Δ*G*_Total binding_**	−67.87 ± 10.34	−102.22 ± 6.73	−99.91 ± 10.58

**Table 5 antibiotics-11-00178-t005:** Total free-binding energies (Δ*G*_Total binding_) and contributing energy terms (Δ*G*_X_) for the *α*-adrenoreceptor inhibitor hits, as well as reference antagonist towards the *C. violaceum* CviR substrate binding site.

Energy (kJ/mol ± SD)	Ligand–Protein Complexes			
	HLC	Comp. 4	Comp. 5	Comp. 6
**Δ*G*_van der Waal_**	−170.53 ± 20.47	−228.70 ± 2.05	−193.04 ± 10.12	−120.62 ± 8.69
**Δ*G*_Electrostatics_**	−29.09 ± 11.43	−84.10 ± 9.68	−46.61 ± 5.18	−52.81 ± 15.96
**Δ*G*_Solvation; Polar_**	153.34 ± 34.91	198.05 ± 11.43	183.12 ± 12.17	120.50 ± 15.88
**Δ*G*_Solvation;_** ** _Apolar; Only-SASA_ **	−20.13 ± 1.36	−23.05 ± 10.19	−22.96 ± 0.84	−18.00 ± 12.27
**Δ*G*_Total binding_**	−66.41 ± 1.65	−137.80 ± 17.97	−79.49 ± 17.95	−70.93 ± 29.05

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
