# Peer review of "Repurposing α-Adrenoreceptor Blockers as Promising Anti-Virulence Agents in Gram-Negative Bacteria"

_antibiotics, 2022, doi:10.3390/antibiotics11020178_

Round 1

Reviewer 1 Report

The present manuscript is a comprehensive evaluation of the possible repurposing of α-Adrenoreceptor Blockers as anti virulence compounds to treat C. violaceum and P. aeruginosa infections.

It is well written and the experiments/results are clear enough to support the utilization of  Terazosin as a virulence inhibitor.

However, I have some major concerns:

  • In the introduction is claimed that antivirulence compounds do not affect bacterial growth and do not select resistance, do not affect normal microbiota and allows the immune system to combat the infections, nevertheless solid experimental evidence demonstrates that this may not be the case:

It is true that antivirulence compounds do not affect bacterial growth of in vitro, in RICH media, however it does so in defined media that has nutrients source which acquisition depends on QS, for example in P. aeruginosa in medium with adenosine or protein as carbon sources.

And indeed resistant mutants against antivirulence compounds such as C-30 furanone had been easily isolated.

See:

Maeda T, García-Contreras R, Pu M, Sheng L, Garcia LR, Tomás M, Wood TK. Quorum quenching quandary: resistance to antivirulence compounds. ISME J. 2012 Mar;6(3):493-501. doi: 10.1038/ismej.2011.122. Epub 2011 Sep 15. PMID: 21918575; PMCID: PMC3280137.

Regarding the claim that QS inhibition do not affect normal microbiota, please either provide examples of the literature demonstrating this or remove it.

For examples demonstrating quorum quenching can severely disrupt microbial communities see:

Waheed H, Xiao Y, Hashmi I, Zhou Y. The selective pressure of quorum quenching on microbial communities in membrane bioreactors. Chemosphere. 2020 May;247:125953. doi: 10.1016/j.chemosphere.2020.125953. Epub 2020 Jan 18. PMID: 32069724.

And for the claim that “virulence inhibition allows the immune system to combat and clear the infections” please consider that usually animal models for the evaluation of virulence inhibition in curing infections are done with immunocompetent animals and it is not clear that infections produced by opportunistic pathogens like P. aeruginosa would be also cleared in immunocompromised animals or humans.

See:

García-Contreras R. Is Quorum Sensing Interference a Viable Alternative to Treat Pseudomonas aeruginosa Infections? Front Microbiol. 2016 Sep 14;7:1454. doi: 10.3389/fmicb.2016.01454. PMID: 27683577; PMCID: PMC5021973.

  • Why QscR was used for docking and not the main QS receptor that hierarchically control QS in P. aeruginosa LasR or RhlR which control QS in low phosphate concentrations,

Soto-Aceves MP, Cocotl-Yañez M, Servín-González L, Soberón-Chávez G. The Rhl Quorum-Sensing System Is at the Top of the Regulatory Hierarchy under Phosphate-Limiting Conditions in Pseudomonas aeruginosa PAO1. J Bacteriol. 2021 Feb 8;203(5):e00475-20. doi: 10.1128/JB.00475-20. PMID: 33288622; PMCID: PMC7890550.

I strongly recomend to implement some docking experiments in these 2 important receptors.

Minor concerns:

  • Some times QS inhibitors do not work for clinical strains, see:

High variability in quorum quenching and growth inhibition by furanone C-30 in Pseudomonas aeruginosa clinical isolates from cystic fibrosis patients

R García-Contreras, B Peréz-Eretza, R Jasso-Chávez, E Lira-Silva, ...

Pathogens and disease 73 (6).

  • Besides your own results, previous experiments show that QS disruption impair the oxidative response in P. aeruginosa and these should be cited:

Huang CT, Shih PC. Effects of quorum sensing signal molecules on the hydrogen peroxide resistance against planktonic Pseudomonas aeruginosa. J Microbiol Immunol Infect. 2000 Sep;33(3):154-8. PMID: 11045377.

 Bjarnsholt T, Jensen PØ, Burmølle M, Hentzer M, Haagensen JAJ, Hougen HP, Calum H, Madsen KG, Moser C, Molin S, Høiby N, Givskov M. Pseudomonas aeruginosa tolerance to tobramycin, hydrogen peroxide and polymorphonuclear leukocytes is quorum-sensing dependent. Microbiology (Reading). 2005 Feb;151(Pt 2):373-383. doi: 10.1099/mic.0.27463-0. PMID: 15699188.

García-Contreras R, Nuñez-López L, Jasso-Chávez R, Kwan BW, Belmont JA, Rangel-Vega A, Maeda T, Wood TK. Quorum sensing enhancement of the stress response promotes resistance to quorum quenching and prevents social cheating. ISME J. 2015 Jan;9(1):115-25. doi: 10.1038/ismej.2014.98. Epub 2014 Jun 17. PMID: 24936763; PMCID: PMC4274435.

  • Please use italics for the scientific names in the references.
  • P 2 L 39 “Professionally,” what do you mean?
  • Please discuss what could be the possible side effects of using α-Adrenoreceptor Blockers as anti virulence compounds.

Author Response

Dear Reviewer,

We are very thankful for your valuable and constructive comments. Please find the reply to the points you raised in the attached file.

Best Regards,

Reviewer 2 Report

The present paper “Repurposing α-Adrenoreceptor Blockers as Promising Anti-Virulence Agents in Gram-Negative Bacteria” reports the in silico studies of some α-blockers on quorum sensing receptors and further in vivo/in vitro studies of the selected derivative; terazosin.

Antibacterial drug resistance is a global health-threatening problem, therefore, drug repurposing can be considered a rational approach to overcome this issue. However, this paper contains the experimental data of only terazosin, so it cannot be claimed that α-adrenoreceptor blockers are promising anti-virulence agents as stated in the title. Because of the additional issues listed below, I do not recommend the publication of this work in Antibiotics with such a high impact factor.

  • The manuscript is too long to be a research paper for such a study. The authors wrote the same things many many times throughout the paper, especially about the docking parts.
  • It is not clear how the authors decided on their molecular docking targets. As they also stated in the paper, there are more molecular targets in QS signaling.
  • The authors started with seven compounds which is not a huge number. So, I find it a wrong approach to eliminate most of the compounds based on just theoretical binding energies after initial docking studies.
  • It would be better to provide the 2D structure of HLC somewhere in the paper for the readers.
  • α-Adrenoreceptor blockers are mainly used for their antihypertensive effects, so I wonder if the authors considered this situation in their study. Using these drugs to achieve anti-virulence agents can cause undesirable blood pressure reduction in healthy people.
  • In the title, the authors claim that α-adrenoreceptor blockers can be considered as promising anti-virulence agents in Gram-negative bacteria, but they actually focused on just P. aeruginosa.

Author Response

(The authors gave the same response as above.)

Reviewer 3 Report

The work is done in a systematic way, where authors performed several experiments of docking, molecular dynamics, in-vitro and in-vivo. However, some queries need to be addressed. 

  1. Page 1, lines 4 & 11; There seems a mistake in the affiliation section ( especially with the affiliation of the corresponding author: Wael A. H. Hegazy as represented with affiliation number 4) is missing, but there is a fifth affiliation with no affiliated author is mentioned. 
  2.  As authors investigated inverse docking tools for repurposing of adrenergic blockers for quorum sensing, therefore, it would be relevant to include a small paragraph about the perspective of inverse docking to attract a wide audience.
  3. Page 28, line 983; Do provide the physiological pH range or exact value for 3D protonation of protein. 
  4. Page 28, line 992; Do provide the reasoning for choosing the active site retrieved from MOE active site finder. As MOE active site finder tool usually provides a list of putative active sites. Therefore, a table with at least the top 3 putative active sites from that list and a tab which provide the necessary reasoning that why those from the list were not selected. For more information, follow these references: (a) doi.org/10.1021/acsomega.7b00010; (b) doi.org/10.1016/j.compbiolchem.2018.02.008; 
  5. Was there any evaluation performed to validate the accuracy or precision of the adopted docking methods that are mentioned in this paper, like self-docking using the same placement method (Triangle matcher) and scoring (GBVI/WSA_dG). for more information, follow these references: (a) MOE assisted repurposing of inhibitors using triangle matcher doi.org/10.1016/j.jsps.2018.01.017; (b) Evaluation of placement methods and others methods of MOE doi.org/10.1016/j.bioorg.2018.12.003 
  6. In computational methods, how false-positive results were differentiated from the considered results? 
  7. Why triangle matcher and London_dG scoring were the only ones selected for the current study when other scoring options were available in MOE?
  8. Do provide the coordinate file of the selected docking pose mentioned in the paper.

The paper is written fluently with some typographical mistakes, which can be revised later. This paper certainly fits in the interest of the current journal.

Author Response

(The authors gave the same response as above.)

Round 2

Reviewer 2 Report

I thank the authors for their replies to my queries about their work. I am sorry but my opinion has not changed. I still think that with so little experimental data, it cannot be claimed that α-adrenoreceptor blockers can be used as promising anti-virulence agents in Gram-negative bacteria. The paper is unnecessarily long to attract the attention of the readers. I suggest the authors experimentally focus on more members of this class of molecules to be so pretentious about the study.